# General framework for online-to-nonconvex conversion: Schedule-free SGD is also effective for nonconvex optimization

**Kwangjun Ahn** [* 1]  **Gagik Magakyan** [* 2]  **Ashok Cutkosky** [3]

## Abstract

This work investigates the effectiveness of schedule-free methods, developed by A. Defazio et al. (NeurIPS 2024), in nonconvex optimization settings, inspired by their remarkable empirical success in training neural networks. Specifically, we show that schedule-free SGD achieves optimal iteration complexity for nonsmooth, nonconvex optimization problems. Our proof begins with the development of a general framework for online-to-nonconvex conversion, which converts a given online learning algorithm into a nonconvex optimization algorithm. Our general framework not only recovers existing conversions but also leads to two novel conversion schemes. Notably, one of these new conversions corresponds directly to schedule-free SGD, allowing us to establish its optimality. Additionally, our analysis provides valuable insights into the parameter choice for schedule-free SGD, addressing a theoretical gap that the convex theory cannot explain.

## 1. Introduction

Training large-scale neural network models, such as large language models, requires a well-designed optimization strategy to ensure stable and fast convergence. For instance, training typically requires a carefully designed optimizer, such as the Adam optimizer (Kingma and Ba, 2014), along with meticulously tuned learning rate scheduling.

Recently, Defazio et al. (2024) introduced the schedule-free method, which achieves impressive training performance

---
[*]Equal contribution  [1]Microsoft Research, Cambridge, MA, USA  [2]Operations Research Center, Massachusetts Institute of Technology, Cambridge, MA, USA  [3]Department of Electrical & Computer Engineering, Boston University, Boston, MA, USA. Correspondence to: Kwangjun Ahn <kwangjunahn@microsoft.com>, Gagik Magakyan <gagmag@mit.edu>, Ashok Cutkosky <ashok@cutkosky.com>.

*Proceedings of the 42nd International Conference on Machine Learning*, Vancouver, Canada. PMLR 267, 2025. Copyright 2025 by the author(s).

without any learning rate scheduling. The schedule-free method is an add-on scheme that can be applied to any chosen base optimizer, converting it into a schedule-free variant. While this method has shown strong empirical performance in training large neural network models, its theoretical analysis has been limited to the convex setting (Defazio et al., 2024). Our aim is to extend the theoretical understanding of schedule-free methods to nonconvex optimization.

As an initial step, this work focuses on the version where the base optimizer is chosen as SGD, referred to as *schedule-free SGD*. For a given learning rate $\gamma > 0$ and interpolation weights $c_t, \kappa_t \in [0, 1]$, the updates of schedule-free SGD maintain $\mathbf{x}_t$, $\mathbf{y}_t$, and $\mathbf{z}_t$, as follows:

$$
\begin{cases}
\mathbf{x}_t = (1 - c_t)\mathbf{x}_{t-1} + c_t\mathbf{z}_t, \\
\mathbf{y}_t = (1 - \kappa_t)\mathbf{z}_t + \kappa_t\mathbf{x}_t, \\
\mathbf{g}_t = \text{a stochastic gradient at } \mathbf{y}_t, \\
\mathbf{z}_{t+1} = \mathbf{z}_t - \gamma\mathbf{g}_t.
\end{cases} \quad \text{(SF-SGD)}
$$

Here, $\mathbf{z}_t$ corresponds to the base SGD trajectory, $\mathbf{x}_t$ is a (weighted) average of $\mathbf{z}_t$, and $\mathbf{y}_t$ is an interpolation between $\mathbf{x}_t$ and $\mathbf{z}_t$ where the stochastic gradient is computed.

### 1.1. Our main result and approach

To understand the effectiveness of schedule-free SGD for training neural networks, we analyze the method in the **nonsmooth and nonconvex** setting (Zhang et al., 2020; Davis et al., 2022b; Tian et al., 2022). Specifically, we adopt the $(\lambda, \epsilon)$-stationarity criterion from (Zhang and Cutkosky, 2024; Ahn and Cutkosky, 2024) (Definition 1), which seeks approximate Goldstein stationary points (Goldstein, 1977). This criterion emerges as a practical framework for analyzing optimization algorithms, particularly considering recent findings that show practical optimizers like SGD with momentum and Adam are theoretically optimal (Zhang and Cutkosky, 2024; Ahn and Cutkosky, 2024).

Our main results demonstrate that schedule-free SGD is optimal not only for convex optimization, as established by Defazio et al. (2024), but also for the most challenging case of nonsmooth and nonconvex optimization. Our main results can be summarized informally as follows:

**Theorem 1.1** (Informal; see Section 5). *Schedule-free SGD* (SF-SGD)*, with an appropriate choice of parameters $\gamma, c_t, \kappa_t$, achieves optimal rates for nonsmooth and nonconvex $F$.*

Our proof technique leverages the online-to-nonconvex conversion framework pioneered by Cutkosky et al. (2023). In essence, this framework takes an online learner as input and outputs an optimization algorithm. As its name suggests, this framework translates online learning guarantees into nonconvex guarantees, analogous to the well-known online-to-batch conversion for convex settings (Cesa-Bianchi et al., 2004). As suggested by our title, we first introduce a general framework for online-to-nonconvex conversion. This framework not only encompasses the previous conversion (Cutkosky et al., 2023; Zhang and Cutkosky, 2024; Ahn and Cutkosky, 2024) as a special case but also enables new conversion schemes, as we present in Section 4.2 and Section 5.

With the new conversion schemes, our main observation is that one of these novel conversions, outlined in Algorithm 5, directly corresponds to schedule-free SGD. Specifically, by choosing a basic online mirror descent as the online learner in Algorithm 5, we naturally recover the schedule-free SGD algorithm. From this, our main result, Theorem 1.1, follows.

Notably, our approach provides fresh practical insight into the parameter choice of schedule-free methods, which cannot be explained by prior convex analysis in Defazio et al. (2024). Our results suggest that setting $\kappa_t$ close to 1 is advantageous for nonconvex optimization. This finding clarifies the curious importance of choosing $\kappa_t$ near 1 in (Defazio et al., 2024) for empirical performance - a phenomenon not previously explained by convex theory.

### 1.2. Related work

Our work builds on a line of research focused on convergence guarantees for nonsmooth, nonconvex optimization. Intuitively, our convergence criterion aims to find approximate Goldstein stationary points (Goldstein, 1977). The formal study of iteration complexity for finding approximate Goldstein stationary points was initiated by Zhang et al. (2020) and has since garnered significant interest (Davis et al., 2022b; Tian and So, 2022; Lin et al., 2022; Chen et al., 2023; Jordan et al., 2023; Cutkosky et al., 2023; Kornowski and Shamir, 2024). Alternative convergence notions are also widely used, including approaches based on the Moreau envelope or by imposing weak convexity conditions (Davis et al., 2018; 2022a). In particular, the criterion we consider in this work, formalized in Definition 1, follows the relaxed version proposed by Zhang and Cutkosky (2024), which slightly modifies the original criterion from Zhang et al. (2020). Additional discussion of related work is provided in Section 7.

## 2. Preliminaries

We first introduce the key assumptions and the notion of convergence. Throughout this paper, unless specified otherwise, $\|\cdot\|$ denotes the $L_2$ norm.

### 2.1. Setting

Following Cutkosky et al. (2023), we consider optimizing a loss function $F$ that satisfies the following conditions.

**Assumption 1.** Let $F : \mathbb{R}^d \to \mathbb{R}$ be a differentiable function with the following properties:

- Let $\Delta_F := F(\mathbf{x}_0) - \inf_{\mathbf{x}} F(\mathbf{x})$.

- For any two points $\mathbf{x}$ and $\mathbf{w}$, $F(\mathbf{x}) - F(\mathbf{w}) = \int_0^1 \langle \nabla F(\mathbf{w} + t(\mathbf{x} - \mathbf{w})), \mathbf{x} - \mathbf{w} \rangle \, \mathrm{d}t$.

- $F$ is $G$-Lipschitz, *i.e.*, for any point $\mathbf{x}$, $\|\nabla F(\mathbf{x})\| \leq G$.

Here, the second condition, called *well-behavedess* in (Cutkosky et al., 2023, Definition 1), is a mild regularity condition. For any locally Lipschitz function $F$, applying an arbitrarily small perturbation is sufficient to ensure this condition (Cutkosky et al., 2023, Proposition 2).

We optimize the loss function $F$ by via a stochastic gradient oracle, which is formalized as follows:

**Assumption 2** (**Stochastic gradient oracle**). We assume access to a stochastic gradient oracle at any point $\mathbf{x}$. More formally, for any given point $\mathbf{x}$, each call to the oracle independently returns a stochastic gradient $\mathbf{g}$ that satisfies the following properties:

$$\mathbb{E}[\mathbf{g}] = \nabla F(\mathbf{x}), \quad \text{and} \quad \mathbb{E}\left[\|\mathbf{g} - \nabla F(\mathbf{x})\|^2\right] \leq \sigma^2.$$

We denote the stochastic gradient oracle as STOGRAD. When the oracle returns the stochastic gradient $\mathbf{g}$ at point $\mathbf{x}$, we write this as $\mathbf{g} \leftarrow$ STOGRAD($\mathbf{x}$).

### 2.2. Approximate Goldstein stationary point

For the notion of optimality, we follow Zhang and Cutkosky (2024); Ahn and Cutkosky (2024) and consider the following notion of stationarity for nonconvex and nonsmooth functions. This notion can be regarded as an approximate version of the notion of a Goldstein stationarity point.

**Definition 1** (($\lambda, \varepsilon$)**-stationary point**). Suppose $F : \mathbb{R}^d \to \mathbb{R}$ is differentiable. We say $\mathbf{x}$ is a ($\lambda, \varepsilon$)-stationary point of $F$ if $\|\nabla F(\mathbf{x})\|^{[\lambda]} \leq \varepsilon$, where

$$\|\nabla F(\mathbf{x})\|^{[\lambda]} := \inf \left\{ \|\mathbb{E}[\nabla F(\mathbf{w})]\| + \lambda \mathbb{E} \|\mathbf{w} - \mathbf{x}\|^2 \right\}.$$

Here the infimum is taken over the distribution $p \in \mathcal{P}(\mathbb{R}^d)$ such that $\mathbb{E}_{\mathbf{w} \sim p}[\mathbf{w}] = \mathbf{x}$.

Our algorithms will identify $(\lambda, \varepsilon)$-stationary points using $\mathcal{O}\left(\lambda^{1/2}\epsilon^{-7/2}\right)$ calls to a stochastic gradient oracle, which is the optimal rate (Zhang and Cutkosky, 2024).

To further motivate this definition, we remark that $(\lambda, \varepsilon)$-stationary points retain the desirable properties of Goldstein stationary points. Specifically, the following result (Zhang and Cutkosky, 2024, Lemma 2.3) demonstrates that, akin to Goldstein stationary points, $(\lambda, \varepsilon)$-stationary points can be reduced to first-order stationary points with appropriate choices of $\lambda$ when the objective function is smooth or second-order smooth.

**Proposition 2.1.** *If $F$ is $L$-smooth, then an $(L^2\varepsilon^{-1}, \varepsilon)$-stationary point $\mathbf{x}$ of $F$ satisfies $\|\nabla F(\mathbf{x})\| \leq 2\varepsilon$. Moreover, if $F$ is $H$-second-order-smooth, then an $(H/2, \varepsilon)$-stationary point $\mathbf{x}$ of $F$ satisfies $\|\nabla F(\mathbf{x})\| \leq 2\varepsilon$.*

Note that another popular notion of approximate Goldstein stationarity is called $(\delta, \varepsilon)$-stationarity due to Zhang et al. (2020). However, the algorithms that achieve optimal complexity under that notion often require clipping on the momentum term (Cutkosky et al., 2023), which introduces deviations from the practical optimization algorithms. Hence, in this work, we adopt the notion of $(\lambda, \varepsilon)$-stationarity, for which it has been demonstrated that optimal algorithms does not require clipping operations (Zhang and Cutkosky, 2024). We also remark that algorithms that identify $(\lambda, \varepsilon)$-stationary points can also identify $(\delta, \varepsilon)$-stationary points when $F$ is Lipschitz, as demonstrated in (Zhang and Cutkosky, 2024, Lemma 2.4).

As mentioned above, the stochastic gradient oracle complexity of finding a $(\lambda, \varepsilon)$ stationary point is $\Theta(\lambda^{1/2}\varepsilon^{-7/2})$ (Zhang and Cutkosky, 2024). By Proposition 2.1, any algorithms achieving this rate (such as the ones we will present), can also find a point $\mathbf{x}$ with $\|\nabla F(\mathbf{x})\| \leq \varepsilon$ in $O(\epsilon^{-4})$ oracle calls when $F$ is smooth and $O(\epsilon^{-3.5})$ calls when $F$ is second-order smooth. These are the optimal rates for their respective function classes (Arjevani et al., 2023; 2020).

### 2.3. Online learning

In this section, we provide a brief background on online learning, which plays a key role in our development. Online learning is modeled as a sequential decision-making process over $T$ rounds. In each round $t$, the online learner chooses a point $\boldsymbol{\delta}_t \in \mathbb{R}^d$, and then a loss function $\ell_t : \mathbb{R}^d \to \mathbb{R}$ is revealed. The learner then incurs a loss $\ell_t(\boldsymbol{\delta}_t)$. The choice of $\boldsymbol{\delta}_t$ is based on the previous loss sequence $\ell_{1:t-1}$, and after selecting $\boldsymbol{\delta}_t$, the learner observes the next loss $\ell_t$.

The performance of the online learner is measured using the *regret* with respect to a comparator $\mathbf{u}$, formally defined as: $\mathsf{Regret}_T(\mathbf{u}) := \sum_{t=1}^{T}(\ell_t(\boldsymbol{\delta}_t) - \ell_t(\mathbf{u}))$. However, recent works (Cutkosky et al., 2023; Ahn et al., 2024) demonstrate that when designing nonconvex optimization algorithms,

base online learners must be adaptive to time-varying comparators, also known as the *dynamic* regret setting. Inspired by Ahn et al. (2024); Zhang and Cutkosky (2024); Ahn et al. (2024), we design such dynamic online learners by considering the following "discounted" version of regret.

**Definition 2 (Discounted regret).** Consider the loss sequence $\ell_{1:T}$. For any $T \geq 1$, we define the discounted regret of an online learner with respect to a comparator $\mathbf{u}$ as:

$$\mathsf{Regret}_T^{[\beta]}(\mathbf{u}) := \sum_{t=1}^{T} \beta^{T-t}\left(\ell_t(\boldsymbol{\delta}_t) - \ell_t(\mathbf{u})\right).$$

For instance, Ahn et al. (2024) demonstrate that online learners with low discounted regret can achieve low dynamic regret through what they call the *discounted-to-dynamic* conversion. Additionally, discounted regret has been shown to be a more effective metric for designing adaptive online learners in dynamic environments across various contexts, including conformal prediction (Zhang et al., 2024) and online linear regression (Jacobsen and Cutkosky, 2024).

## 3. General framework for online-to-nonconvex conversion

In this section, we introduce a general scheme (Algorithm 1) for converting online learning guarantees into nonconvex optimization guarantees. As a preview, we will demonstrate that our Algorithm 1 not only recovers existing approaches as special cases, but also enables the design of novel conversion methods. Let us begin with the pseudocode.

---

**Algorithm 1** General scheme for online-to-nonconvex conversion

1: **Input:** Initial iterates $\mathbf{x}_0 = \mathbf{w}_0$, an online learner $\mathcal{A}$, and $T \in \mathbb{N}$, regularization strength $\mu \geq 0$.
2: **for** $t = 1, 2, \ldots, T$ **do**
3:     Receive $\boldsymbol{\delta}_t$ from $\mathcal{A}$.
4:     Choose $\mathbf{x}_t$ arbitrarily. **// The design choice**
5:     Update $\mathbf{w}_t = \mathbf{x}_t + \boldsymbol{\delta}_t$.
6:     Update $\mathbf{y}_t = \mathbf{x}_t + s_t\boldsymbol{\delta}_t$ where $0 \leq s_t \leq 1$ is drawn uniformly *i.i.d.*
7:     Compute $\mathbf{g}_t \leftarrow \textsc{StoGrad}(\mathbf{y}_t)$.
8:     Send loss $\ell_t(\cdot) = \langle \mathbf{g}_t, \cdot \rangle + \frac{\mu}{2}\|\cdot\|^2$ to $\mathcal{A}$.
9: **end for**

---

Overall, Algorithm 1 generates three sequences of iterates: $\mathbf{x}_t$, $\mathbf{y}_t$, and $\mathbf{w}_t$. At each iteration, the stochastic gradients are fed into the online learner, which outputs $\boldsymbol{\delta}_t$.

The main design feature of Algorithm 1 that gives it flexibility is the ability to choose $\mathbf{x}_t$ arbitrarily at each iteration. However, there are technical properties we want $\mathbf{x}_t$ to sat-

isfy in order to achieve better nonconvex guarantees, which are detailed in Section 3.3.

The online learner's output $\boldsymbol{\delta}_t$ is then used alongside $\mathbf{x}_t$ to compute $\mathbf{w}_t$ according to the update rule $\mathbf{w}_t = \mathbf{x}_t + \boldsymbol{\delta}_t$. Additionally, the iterates $\mathbf{y}_t$ are sampled uniformly from the line segment connecting $\mathbf{x}_t$ and $\mathbf{w}_t$, and the stochastic gradients are computed at these $\mathbf{y}_t$ iterates.

### 3.1. Output of the general scheme

To establish nonconvex optimization guarantees, we utilize the exponential moving average (EMA) of the $\mathbf{y}_t$ iterates, similar to (Zhang and Cutkosky, 2024; Ahn and Cutkosky, 2024). We begin by formally defining these EMA iterates.

**Definition 3.** Given a discount factor $\beta \in (0,1)$ and a sequence of iterates $\{\mathbf{y}_s\}_{s=1}^t$, the $\beta$-EMA of the sequence up to time $t$, denoted as $\overline{\mathbf{y}}_t$, is defined as

$$\overline{\mathbf{y}}_t = \frac{1-\beta}{1-\beta^t} \sum_{s=1}^t \beta^{t-s} \mathbf{y}_s.$$

In particular, as we will see in Lemma 3.1, the final output will be the random EMA iterate $\overline{\mathbf{y}}_\tau$, where $\tau$ is a carefully selected random index, defined as follows.

**Definition 4** (Random index distribution). Let $\tau$ be a random index distributed over $\{1, 2, \ldots, T\}$ with the following distribution:

$$\Pr(\tau = t) = \begin{cases} \frac{1-\beta^t}{T}, & \text{for } t = 1, \ldots, T-1, \\ \frac{1}{1-\beta} \cdot \frac{1-\beta^T}{T}, & \text{for } t = T. \end{cases}$$

We note that previous works select the output uniformly at random from the sequence $\{\overline{\mathbf{y}}_t\}_{t=1}^T$ (Zhang and Cutkosky, 2024; Ahn and Cutkosky, 2024). Our carefully designed random EMA iterate from Definition 4 leads to an improved conversion result, offering stronger nonconvex optimization guarantees, as we will discuss in Section 4.

Notice that as $\beta$ approaches 1, the random index $\tau$ assigns a significantly higher probability (by a multiplicative factor of $\frac{1}{1-\beta}$) to the final index $T$. This aligns with common practice, where the final iterate—rather than an average of previous iterates—is often used as the output. Indeed, we will select $\beta$ very close to 1 for our nonconvex guarantees (we will choose $\beta = 1 - O(\varepsilon^2)$).

### 3.2. Conversion guarantees

Given the description of the algorithm and its output, we now present the online-to-nonconvex conversion guarantees. We begin by introducing a key definition that quantifies the stability of the $\mathbf{x}_t$ sequence.

**Definition 5** (x-iterate stability). Consider the iterates generated by Algorithm 1. The iterate stability factor, denoted

by $C_{\mathbf{x}} \geq 0$, is the smallest nonnegative constant such that:

$$\mathbb{E} \sum_{t=1}^T \|\mathbf{x}_t - \mathbf{x}_{t-1}\|^2 \leq C_{\mathbf{x}} \cdot \mathbb{E} \sum_{t=1}^T \|\boldsymbol{\delta}_t\|^2.$$

With the concept of the iterate stability factor, we can now state the conversion result.

**Lemma 3.1** (Generic online-to-nonconvex conversion). *Consider the iterates generated according to Algorithm 1. For $\beta \in (0,1)$ and $D > 0$, define the comparators for online learner as follows:*

$$\forall t \in [T], \quad \mathbf{u}_t := -D \frac{\sum_{s=1}^t \beta^{t-s} \nabla F(\mathbf{y}_s)}{\left\| \sum_{s=1}^t \beta^{t-s} \nabla F(\mathbf{y}_s) \right\|}.$$

*Then, as long as the regularization strength satisfies $\mu \geq 8\lambda D \left(1 + C_{\mathbf{x}}(1-\beta)^{-2}\right)$, the following holds:*

$$\begin{aligned}
\mathbb{E}_\tau \|\nabla F(\overline{\mathbf{y}}_\tau)\|^{[\lambda]} &\leq \frac{\beta}{DT} \mathbb{E}\left[ \mathsf{Regret}_T^{[\beta]}(\mathbf{u}_T) \right] \\
&+ \frac{1-\beta}{DT} \sum_{t=1}^T \mathbb{E}\left[ \mathsf{Regret}_t^{[\beta]}(\mathbf{u}_t) \right] \\
&+ \frac{1}{DT} \mathbb{E} \sum_{t=1}^T (F(\mathbf{x}_t) - F(\mathbf{w}_t)) + \frac{\mu D}{2} \\
&+ \frac{\sigma}{T\sqrt{1-\beta}} + \sigma\sqrt{1-\beta}.
\end{aligned}$$

*Proof.* See Appendix A. $\square$

Lemma 3.1 provides an upper bound on the main quantity of interest, $\mathbb{E}_\tau \|\nabla F(\overline{\mathbf{y}}_\tau)\|^{[\lambda]}$. One of the main terms in this upper bound is the sum of the discounted regret terms, which indicates that improved performance by the online learner leads to better guarantees in nonconvex optimization. Thus, Lemma 3.1 effectively translates the online learning guarantee into a nonconvex optimization guarantee, as suggested by its name, *online-to-nonconvex* conversion.

However, in its current form, the presence of several additional terms in the upper bound makes the result less interpretable. Before demonstrating the strength of this general conversion framework, we first reformulate Lemma 3.1 into a more interpretable and user-friendly form.

### 3.3. User-friendly nonconvex optimization guarantees

In this section, we apply a concrete discounted regret bound to Lemma 3.1 to derive a more user-friendly nonconvex guarantee, which will be used throughout the remainder of the paper. In particular, a discounted version of composite objective online mirror descent (OMD) (Beck and Teboulle, 2003; Duchi et al., 2010; Zhang and Cutkosky, 2024) achieves the following discounted regret bound.

**Lemma 3.2.** *Let $\beta \in (0, 1)$, $\mu \geq 0$, $\eta > 0$, and a sequence of vectors $\{\mathbf{g}_t\}_{t=1}^T$. Suppose that $\mathbb{E} \|\mathbf{g}_t\|^2 \leq G^2 + \sigma^2$ for all $t \in [T]$. Consider an online learner initialized with $\boldsymbol{\delta}_1 = \mathbf{0}$ and updated as follows:*

$$\boldsymbol{\delta}_{t+1} = \frac{\beta}{1 + \eta\mu} \left(\boldsymbol{\delta}_t - \eta\mathbf{g}_t\right). \qquad (\beta\text{-OMD})$$

*Then, this online learner with $\eta = \frac{2}{G+\sigma} \|\mathbf{u}\| \sqrt{1 - \beta}$ achieves the following discounted regret bound:*

$$\mathbb{E}\left[\mathsf{Regret}_T^{[\beta]}(\mathbf{u})\right] \leq \frac{2 \|\mathbf{u}\| (G + \sigma)}{\beta\sqrt{1 - \beta}} + \frac{\mu}{2} \|\mathbf{u}\|^2. \quad (1)$$

*Proof.* See Section D.1. □

We expect that alternative online learning frameworks, such as follow-the-regularized-leader, can achieve the discounted regret bound similar to (1). However, a comprehensive exploration of discounted online learners lies outside the scope of this work.

Using the bound (1), we can derive an user-friendly nonconvex guarantee as follows. In the following two sections, only the regret bound (1) is important; the specifics of the discounted OMD algorithm are irrelevant. Any algorithm achieving a similar (or better) regret bound would provide the same (or improved) results.

**Theorem 3.1** (**Generic nonconvex guarantees**). *Consider the iterates generated according to Algorithm 1, where the online learner $\mathcal{A}$ achieves the discounted regret given by (1). Let $\varepsilon > 0$ be such that $\varepsilon \leq \frac{7}{2}(G + \sigma)$ and let $T \geq 49(G + \sigma)^2 \varepsilon^{-2}$. Then, there exists a choice of parameters $\beta = \beta_\star$, $D = D_\star$, $\mu = \mu_\star$ such that the following holds:*

$$\mathbb{E}_\tau \|\nabla F(\overline{\mathbf{y}}_\tau)\|^{[\lambda]} \leq 3\varepsilon + \frac{4\lambda^{1/2}\varepsilon^{-1/2}\Gamma_{\mathbf{x}}}{T}.$$

*Here the algorithm-dependent quantity $\Gamma_{\mathbf{x}}$ is defined as*

$$\boxed{\Gamma_{\mathbf{x}} := \left(1 + \frac{49(G + \sigma)^2 \sqrt{C_{\mathbf{x}}}}{\varepsilon^2}\right) \mathbb{E} \sum_{t=1}^T (F(\mathbf{x}_t) - F(\mathbf{w}_t))}$$

*and the iterate stability factor $C_{\mathbf{x}}$ is defined in Definition 5.*

*Proof.* See Appendix B. □

The key takeaway is that the nonconvex guarantees are determined by the magnitude of $\Gamma_{\mathbf{x}}$. Thus, it is essential to select the iterates $\mathbf{x}_t$ in a way that minimizes $\Gamma_{\mathbf{x}}$. As a warm-up, we examine two simple special cases that lead to optimal complexity.

# 4. Warm-up: two simple ways to achieve optimal nonconvex guarantees

In this section, as a warm-up, we present two simple ways to achieve the optimal nonconvex guarantee using our general conversion scheme, Algorithm 1. Since Algorithm 1 maintains two sequences of iterates, $\mathbf{x}_t$ and $\mathbf{w}_t$, perhaps the two simplest options for $\mathbf{x}_t$ are as follows:

I. Set $\mathbf{x}_t = \mathbf{w}_{t-1}$ for all $t \in [T]$.

II. Set $\mathbf{x}_t = \mathbf{x}_{t-1}$ for all $t \in [T]$.

As we will demonstrate shortly, the first option recovers the previous conversion, while the second option leads to a novel conversion scheme, showcasing the versatility of our general framework. We will build on these warm-up cases to analyze schedule-free SGD in Section 5. Throughout this section, $\mathcal{A}$ can be any online learner that achieves the discounted regret bound (1).

## 4.1. Option I leads to previous conversion

We begin with the first option, as described in Algorithm 2.

---
**Algorithm 2** Option I
1: In Algorithm 1, set $\mathbf{x}_t = \mathbf{w}_{t-1}$ for all $t \in [T]$.
---

Algorithm 2 recovers previous approaches (Cutkosky et al., 2023; Zhang and Cutkosky, 2024; Ahn and Cutkosky, 2024). In particular, since Algorithm 2 leads to the update $\mathbf{w}_t - \mathbf{w}_{t-1} = \boldsymbol{\delta}_t$, it provides a nice interpretation of selecting the increments $\mathbf{w}_t - \mathbf{w}_{t-1}$ based on the online learner's output, as highlighted by (Ahn et al., 2024).

The main advantage of Algorithm 2 is that the cumulative sum of loss decrements can be kept small due to a telescoping sum (recall that $\Delta_F := F(\mathbf{x}_0) - \inf_{\mathbf{x}} F(\mathbf{x})$):

$$\sum_{t=1}^T (F(\mathbf{x}_t) - F(\mathbf{w}_t)) = \sum_{t=1}^T (F(\mathbf{w}_{t-1}) - F(\mathbf{w}_t))$$
$$= F(\mathbf{w}_0) - F(\mathbf{w}_T) \leq \Delta_F. \qquad (2)$$

Moreover, from Algorithm 2, we have $\|\mathbf{x}_t - \mathbf{x}_{t-1}\| = \|\boldsymbol{\delta}_{t-1}\|$ for $t \geq 2$, and for $t = 1$, $\|\mathbf{x}_t - \mathbf{x}_{t-1}\| = \|\mathbf{w}_0 - \mathbf{x}_0\| = 0$. Therefore, we have

$$\mathbb{E} \sum_{t=1}^T \|\mathbf{x}_t - \mathbf{x}_{t-1}\|^2 \leq \mathbb{E} \sum_{t=1}^T \|\boldsymbol{\delta}_t\|^2, \qquad (3)$$

which shows that the iterate stability factor $C_{\mathbf{x}}$ is at most 1.

Combining these two calculations, Theorem 3.1 leads to the following nonconvex optimization guarantee.

**Corollary 4.1.** *Consider the iterates of Algorithm 2. Under the setting of Theorem 3.1, it holds that $\mathbb{E}_\tau \|\nabla F(\overline{\mathbf{y}}_\tau)\|^{[\lambda]} \leq 4\varepsilon$, provided that*

$$T \geq 49(G+\sigma)^2 \varepsilon^{-2} \cdot \max\left\{8\Delta_F \lambda^{1/2} \varepsilon^{-3/2}, \ 1\right\}. \quad (4)$$

*Proof.* With Algorithm 2, the above inequalities, (2) and (3), show that $\sum_{t=1}^T (F(\mathbf{x}_t) - F(\mathbf{w}_t)) \leq \Delta_F$ and $C_{\mathbf{x}} \leq 1$. Therefore, it follows that

$$\Gamma_{\mathbf{x}} \leq \left(1 + \frac{49(G+\sigma)^2}{\varepsilon^2}\right) \Delta_F \leq 2 \cdot \frac{49(G+\sigma)^2}{\varepsilon^2} \Delta_F,$$

since $\varepsilon \leq \frac{7}{2}(G+\sigma)$. Thus, Theorem 3.1 yields:

$$\mathbb{E}_\tau \left[\|\nabla F(\overline{\mathbf{y}}_\tau)\|^{[\lambda]}\right] \leq 3\varepsilon + \frac{8\lambda^{1/2}\varepsilon^{-1/2}\left(\frac{49(G+\sigma)^2}{\varepsilon^2}\Delta_F\right)}{T},$$

provided that $T \geq 49(G+\sigma)^2 \varepsilon^{-2}$. The second term in the upper bound is at most $\varepsilon$ if $T \geq 49(G+\sigma)^2 \cdot 8\Delta_F \lambda^{1/2} \varepsilon^{-7/2}$. Hence taking the maximum of the two requirments for $T$, the iteration complexity bound (4) follows. $\square$

We note that the complexity bound in Corollary 4.1 is optimal, in light of the lower bound results of (Zhang and Cutkosky, 2024). In fact, our use of the random index $\tau$ leads to a slightly better guarantee; the upper bound in (Zhang and Cutkosky, 2024) has the second argument of the maximum as $\mathcal{O}\left((G+\sigma)\varepsilon^{-1}\right)$, whereas ours is $\mathcal{O}(1)$.

We emphasize here that Algorithm 2 recovers commonly used momentum-based optimizers as special cases. Zhang and Cutkosky (2024) show that setting $\mathcal{A}$ as the OMD in Lemma 3.2 corresponds to SGD with momentum, while Ahn and Cutkosky (2024) demonstrate that choosing $\mathcal{A}$ as a discounted version of FTRL results in the Adam optimizer, up to minor modifications.

**4.2. Option II leads to a novel optimal conversion**

We now consider the second option. Since $\mathbf{x}_t = \mathbf{x}_{t-1}$ for all $t$, it follows that $\mathbf{x}_t \equiv \mathbf{x}_0$ for all $t \in [T]$.

---

**Algorithm 3** Option II
1: In Algorithm 1, set $\mathbf{x}_t = \mathbf{x}_0$ for all $t \in [T]$.

---

The main advantage of Algorithm 3 is that $C_{\mathbf{x}} = 0$ because $\mathbf{x}_t$ is fixed for all iterates. This implies that

$$\Gamma_{\mathbf{x}} = \mathbb{E}\sum_{t=1}^T (F(\mathbf{x}_0) - F(\mathbf{w}_t)). \quad (5)$$

Hence, we only need to control the term (5), which can be done via the following anchoring scheme.

---

**Algorithm 4** Anchoring scheme
1: **Input:** Initial iterate $\mathbf{x}_0$, and integers $N, T \in \mathbb{N}$.
2: Set the initial anchor point $\mathbf{a}_1 := \mathbf{x}_0$.
3: **for** $n = 1, 2, \ldots, N$ **do**
4:     Starting from $\mathbf{a}_n$, run Algorithm 3 for $T$ iterations to generate the iterates $\{\mathbf{x}_t^{(n)}, \mathbf{w}_t^{(n)}, \mathbf{y}_t^{(n)}\}_{t=1}^T$. (By the choice in Algorithm 3, we have $\mathbf{x}_t^{(n)} \equiv \mathbf{a}_n$ for all $t \in [T]$.)
5:     Sample the next anchor point $\mathbf{a}_{n+1}$ uniformly at random from $\{\mathbf{w}_t^{(n)}\}_{t=1}^T$.
6: **end for**

---

**Corollary 4.2.** *Consider the iterates of Algorithm 4. Under the setting of Theorem 3.1, it holds that*

$$\mathbb{E}_{n \sim \mathrm{Unif}([N])} \mathbb{E}_\tau \left\|\nabla F(\overline{\mathbf{y}}_\tau^{(n)})\right\|^{[\lambda]} \leq 4\varepsilon, \quad (6)$$

*provided that $N \geq 4\Delta_F \lambda^{1/2} \varepsilon^{-3/2}$ and $T \geq 49(G+\sigma)^2 \varepsilon^{-2}$. Hence, the total iteration complexity is:*

$$T \cdot N \geq 49(G+\sigma)^2 \varepsilon^{-2} \cdot \max\left\{4\Delta_F \lambda^{1/2} \varepsilon^{-3/2}, \ 1\right\}.$$

*Proof.* By Theorem 3.1 together with (5), the following holds for each epoch $n$:

$$\mathbb{E}_\tau \left\|\nabla F(\overline{\mathbf{y}}_\tau^{(n)})\right\|^{[\lambda]}$$
$$\leq 3\varepsilon + \frac{4\lambda^{1/2}\varepsilon^{-1/2}}{T}\mathbb{E}\sum_{t=1}^T \left(F(\mathbf{a}_n) - F(\mathbf{w}_t^{(n)})\right)$$
$$= 3\varepsilon + 4\lambda^{1/2}\varepsilon^{-1/2}\mathbb{E}\left(F(\mathbf{a}_n) - F(\mathbf{a}_{n+1})\right),$$

where the last equality follows because $\mathbf{a}_{n+1}$ is chosen uniformly at random from $\{\mathbf{w}_t^{(n)}\}_{t=1}^T$.

Summing over all epochs, since $\sum_{n=1}^N (F(\mathbf{a}_n) - F(\mathbf{a}_{n+1})) = F(\mathbf{a}_1) - F(\mathbf{a}_{N+1})$, we have:

$$\mathbb{E}_{n \sim \mathrm{Unif}([N])} \mathbb{E}_\tau \left\|\nabla F(\overline{\mathbf{y}}_\tau^{(n)})\right\|^{[\lambda]}$$
$$\leq 3\varepsilon + \frac{4\lambda^{1/2}\varepsilon^{-1/2}}{N}\mathbb{E}\left(F(\mathbf{a}_1) - F(\mathbf{a}_{N+1})\right)$$
$$\leq 3\varepsilon + \frac{4\lambda^{1/2}\varepsilon^{-1/2}\Delta_F}{N}.$$

Thus, setting $N \geq 4\Delta_F \lambda^{1/2} \varepsilon^{-3/2}$ ensures that the right-hand side is at most $4\varepsilon$. $\square$

This result demonstrates that Algorithm 4 provides an alternative approach for achieving the optimal nonconvex guarantee. Interestingly, this method bears a conceptual resemblance to the classic online-to-convex conversion (Cesa-Bianchi et al., 2004). In particular, the traditional conversion

runs an online learner for $T$ iterations and then selects an iterate uniformly at random (or averages iterates, applying Jensen's inequality). This approach closely parallels the procedure of a single epoch of Algorithm 3. This is analogous to non-convex optimization approaches based upon repeatedly solving convex subproblems created by appropriate regularization, as discussed by Chen and Hazan (2024).

Thus far, we explored two simple methods for selecting $\mathbf{x}_t$, both of which can lead to the optimal nonconvex guarantee. In the next section, we will consider yet another conversion approach that leads to the optimal guarantee.

# 5. Schedule-free SGD is effective for nonconvex optimization

In this section, we build on Section 4 and consider another special case of Algorithm 1 that achieves the optimal nonconvex guarantee. Specifically, we fix the online learner $\mathcal{A}$ to be $\beta$-OMD from Lemma 3.2. This is in contrast to the previous sections in which the specifics of the online learner were not important. Recall the update rule of $\beta$-OMD:

$$\boxed{\boldsymbol{\delta}_{t+1} = \zeta \left( \boldsymbol{\delta}_t - \eta \mathbf{g}_t \right).} \qquad (\beta\text{-OMD})$$

where we let $\zeta := \frac{\beta}{1+\eta\mu}$ to simplify.

## 5.1. Yet another optimal conversion

Consider the following special case of Algorithm 1, specifically designed for $\beta$-OMD.

---
**Algorithm 5** Option III
---
1: In Algorithm 1 with $\mathcal{A}$ chosen as $\beta$-OMD, choose $\mathbf{x}_t = \mathbf{x}_{t-1} + \frac{1}{\zeta}\boldsymbol{\delta}_t$ for all $t \in [T]$.
---

We first demonstrate that this conversion scheme achieves the optimal nonconvex guarantee. The key idea is that, with Algorithm 5, the iterates $\mathbf{x}_t$ remain sufficiently close to $\mathbf{w}_{t-1}$, allowing us to take advantage of the telescoping sum from Algorithm 2. More specifically, with Algorithm 5, one can easily check that the following holds:

$$\mathbf{x}_t - \mathbf{w}_{t-1} = \frac{1}{\zeta}\boldsymbol{\delta}_t - \boldsymbol{\delta}_{t-1} = -\eta\mathbf{g}_{t-1}. \qquad (7)$$

In other words, $\|\mathbf{x}_t - \mathbf{w}_{t-1}\|$ is significantly smaller than the size of the update made by $\mathcal{A}$. With this observation, we can now show that Algorithm 5 achieves the optimal nonconvex optimization guarantee.

**Corollary 5.1.** *Consider the iterates of Algorithm 5 with the parameter choices as in Theorem 3.1, i.e., $\beta = \beta_\star$, $D = D_\star$, $\mu = \mu_\star$, and the parameters of $\beta$-OMD chosen as:*

$$\eta = \eta_\star := \frac{2}{G+\sigma}D_\star\sqrt{1-\beta_\star} \quad \text{and} \quad \zeta = \zeta_\star := \frac{\beta_\star}{1+\eta_\star\mu_\star}.$$

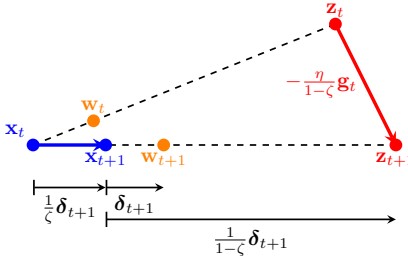

*Figure 1.* Illustration of how Algorithm 5 can be interpreted as schedule-free SGD. By defining the $\mathbf{z}$-iterates according to (8), it becomes clear that the $\mathbf{z}$-iterates follow the base SGD trajectory of schedule-free SGD (SF-SGD).

*Then, it holds that $\mathbb{E}_\tau \|\nabla F(\overline{\mathbf{y}}_\tau)\|^{[\lambda]} \leq 5\varepsilon$, provided that*

$$T \geq 49(G+\sigma)^2\varepsilon^{-2} \cdot \max\left\{20\Delta_F\lambda^{1/2}\varepsilon^{-3/2}, \ 1\right\}.$$

*Proof.* The proof is very similar to that of Corollary 4.1; see Appendix C. □

Corollary 5.1 shows that Algorithm 5 achieves the same optimal complexity as the conversion methods discussed in Section 4, differing only by multiplicative constant factors. Next, we will examine the update rule more closely.

## 5.2. Option III is equivalent to schedule-free SGD

A striking outcome of our general conversion framework (Algorithm 1) is that one of its special cases, Algorithm 5, turns out to be equivalent to the schedule-free SGD method.

To demonstrate this equivalence, we introduce a set of extrapolated iterates, $\mathbf{z}_t$, defined as follows:

$$\mathbf{z}_t := \mathbf{x}_t + \frac{1}{1-\zeta}\boldsymbol{\delta}_t. \qquad (8)$$

These $\mathbf{z}_t$ iterates follow an SGD trajectory, just like the base method used in schedule-free SGD, with an effective step size of $\gamma = \frac{\eta}{1-\zeta}$. See Section D.2 for details.

**Proposition 5.1.** *Under Algorithm 5 and using the extrapolated iterates defined in (8), the following holds:*

$$\mathbf{z}_{t+1} - \mathbf{z}_t = -\frac{\eta}{1-\zeta}\mathbf{g}_t.$$

Next, we demonstrate that Algorithm 5 mirrors the update rule of schedule-free SGD. We begin by presenting the explicit form of Algorithm 5, where we substitute the choice of $\mathbf{x}_t$ according to Algorithm 5, *i.e.*, $\mathbf{x}_t = \mathbf{x}_{t-1} + \frac{1}{\zeta}\boldsymbol{\delta}_t$, along with the choice of $\mathcal{A}$ as $\beta$-OMD. The resulting explicit update form is shown in Algorithm 6.

---

**Algorithm 6** Explicit form of Algorithm 5

1: **for** $t = 1, 2, \ldots, T$ **do**
2:     Receive $\boldsymbol{\delta}_t$ from $\mathcal{A} = \beta$-OMD, *i.e.*, $\boldsymbol{\delta}_t = \zeta(\boldsymbol{\delta}_{t-1} - \eta \mathbf{g}_{t-1})$.
3:     Update $\mathbf{x}_t = \mathbf{x}_{t-1} + \frac{1}{\zeta} \boldsymbol{\delta}_t$.
4:     Update $\mathbf{w}_t = \mathbf{x}_t + \boldsymbol{\delta}_t$.
5:     Set $\mathbf{y}_t = \mathbf{x}_t + s_t \boldsymbol{\delta}_t$, where $s_t$ is drawn uniformly from $[0,1]$ *i.i.d.*
6:     Compute $\mathbf{g}_t \leftarrow \text{STOGRAD}(\mathbf{y}_t)$.
7:     Send loss $\ell_t(\cdot) = \langle \mathbf{g}_t, \cdot \rangle + \frac{\mu}{2} \|\cdot\|^2$ to $\mathcal{A}$.
8: **end for**

---

**Algorithm 7** Rewriting of Algorithm 5 using the extrapolated $\mathbf{z}$-iterates (8)

1: **Input:** Initial iterates $\mathbf{x}_0 = \mathbf{z}_0$.
2: **for** $t = 1, 2, \ldots, T$ **do**
3:     Update $\mathbf{x}_t = \zeta \mathbf{x}_{t-1} + (1 - \zeta) \mathbf{z}_t$.
4:     Set $\mathbf{y}_t = \kappa_t \mathbf{x}_t + (1 - \kappa_t) \mathbf{z}_t$, where $\kappa_t$ is drawn uniformly from $[\zeta, 1]$, *i.i.d.*
5:     Compute $\mathbf{g}_t \leftarrow \text{STOGRAD}(\mathbf{y}_t)$.
6:     Update $\mathbf{z}_{t+1} = \mathbf{z}_t - \gamma \mathbf{g}_t$, where the step size is chosen as $\gamma = \frac{\eta}{1 - \zeta}$.
7: **end for**

---

We can observe that Algorithm 6 can be reformulated in terms of the iterates $\mathbf{x}_t$, $\mathbf{y}_t$, and $\mathbf{z}_t$, eliminating the dependence on $\mathbf{w}_t$. This follows from (8), along with the choice $\mathbf{x}_t = \mathbf{x}_{t-1} + \frac{1}{\zeta} \boldsymbol{\delta}_t$, which implies the following relationship:

$$\mathbf{x}_t = \zeta \mathbf{x}_{t-1} + (1 - \zeta) \mathbf{z}_t.$$

This result is also visually illustrated in Figure 1. Additionally, since $\mathbf{y}_t = \mathbf{x}_t + s_t \boldsymbol{\delta}_t$, we have:

$$\begin{aligned} \mathbf{y}_t &= \mathbf{x}_t + s_t(1 - \zeta)(\mathbf{z}_t - \mathbf{x}_t) \\ &= \big(1 - s_t(1 - \zeta)\big)\mathbf{x}_t + s_t(1 - \zeta)\mathbf{z}_t. \end{aligned}$$

By combining these steps, we obtain Algorithm 7, a reformulation of Algorithm 6. Notably, this algorithm is equivalent to the schedule-free SGD method. Specifically, Algorithm 7 selects $\kappa_t$ uniformly from the interval $[\zeta, 1]$ at each iteration, employs a step size $\gamma = \frac{\eta}{1 - \zeta}$, and consistently sets $c_t \equiv 1 - \zeta$ for all $t$.

### 5.3. Practical insights from our results

Our results highlight an important property of schedule-free SGD: it not only achieves the optimal convex guarantee established by Defazio et al. (2024) but also attains the optimal nonconvex guarantee. This versatility helps explain the empirical success of schedule-free methods across a broad spectrum of optimization problems. It is important to note that our current analysis requires distinct parameter settings for $\gamma$ and $c_t$, depending on whether $F$ is convex or nonconvex.

We now discuss how our results offer new insight into parameter selection for schedule-free SGD. To begin, we note the following fact (see Section D.3 for further details).

**Proposition 5.2.** *With the parameter choices given in Corollary 5.1, we have $\zeta = \zeta_\star = 1 - \Theta\left(\frac{\varepsilon^2}{(G+\sigma)^2}\right)$. Specifically, the parameter $\kappa_t$ in schedule-free SGD is chosen uniformly from $[\zeta_\star, 1]$, implying that $1 - \Theta\left(\frac{\varepsilon^2}{(G+\sigma)^2}\right) \leq \kappa_t \leq 1$ for all $t$. This selection ensures that $\kappa_t$ remains close to 1.*

We interpret these parameter choice in light of empirical findings by Defazio et al. (2024) that lacked theoretical explanation. Their convex guarantee (Defazio et al., 2024, Theorem 2) permits $\kappa_t$ to be chosen arbitrarily within the interval $[0, 1]$, yet experimental results indicated that selecting $\kappa_t$ near 1 (e.g., 0.98) was crucial for strong empirical performance. Our results provides theoretical support for this choice, as noted in the first bullet point of Proposition 5.2.

## 6. Conclusion and future directions

Motivated by the impressive empirical performance of schedule-free methods, this work investigates their effectiveness for nonconvex optimization. As a first step, we demonstrate that schedule-free SGD achieves optimal iteration complexity for nonsmooth, nonconvex optimization. This is accomplished through a general conversion framework that not only recovers existing conversions but also introduces two novel conversion schemes. Notably, one of these novel conversions directly corresponds to schedule-free SGD, which serves as the basis for our analysis.

While this paper lays important groundwork, it merely scratches the surface of our understanding of schedule-free methods and opens up several avenues for future research. Below, we outline a few of these potential directions for the reader's interest.

**Other special cases of our general conversion.** In this work, we explore three specific instances of the general conversion framework. However, it is unlikely that these are the only viable conversions. Since these special cases yield highly practical optimizers, it would be worthwhile to investigate additional special cases and assess the practical implications of those conversions.

**Adaptive schedule-free methods.** Considering the impressive practical performance of the schedule-free version of Adam, as highlighted in (Defazio et al., 2024), it would be intriguing to explore whether this method can be under-

stood as a special case of the general online-to-nonconvex conversion framework. Our current analysis provides a correspondence only for schedule-free SGD.

**Advanced weighting schemes.** The convex analysis of (Defazio et al., 2024) allows for an arbitrary sequence of "weights" for each example which inform the choice for $c_t$ in (SF-SGD). While uniform weighting (corresponding to $c_t = 1/t$) is worst-case optimal, it is not *instance* optimal, and in fact certain empirical heuristics such as learning rate warmup can be recovered by instance-dependent weighting (Defazio et al., 2023). Our analysis makes use of exponentially increasing weights, corresponding to a consant $c_t = \gamma$ as detailed in Algorithm 7. While our weights achieve the optimal worst-case convergence guarantees, we conjecture that improvements are possible by incorporating instance-dependent weighting.

**Truly universal methods.** It is noteworthy that the schedule-free method achieves optimal rates for nonsmooth losses regardless of their convexity. However, as discussed in Section 5.3, the current analysis requires distinct parameter settings for $\gamma$ and $c_t$, depending on whether $F$ is convex or nonconvex. This indicates that we have not yet developed a fully unified algorithm that seamlessly addresses both cases. Nevertheless, these findings suggest the potential for a unified algorithmic framework. It would be valuable to explore whether a single parameter choice can be effective for both scenarios and to determine if such choices align with those observed in practice.

**Why is schedule-free schedule-free?** Lastly, we acknowledge that the main limitation of our analysis is its inability to fully explain why schedule-free methods can effectively alleviate the need for learning rate decay schedules, which is their most notable empirical advantage. Addressing this gap likely requires the development of a theoretical framework for understanding learning rate scheduling in nonconvex optimization.

## 7. Additional related work

As discussed in Section 1.1, our approach specifically builds on leverages the online-to-nonconvex conversion framework introduced by Cutkosky et al. (2023), a versatile framework that established the first optimal algorithm for stochastic nonsmooth, nonconvex optimization. Similar to the well-known online-to-batch conversion (Cesa-Bianchi et al., 2004), which transforms an online learning algorithm into an optimization algorithm for convex losses, this framework adapts this transformation to nonconvex settings. A key distinction, as noted by Ahn et al. (2024), lies in how each framework sets the iterates of the resulting optimization algorithm. While the online-to-batch conversion sets the

iterates, $\mathbf{x}_t$, directly according to the online learner, the online-to-nonconvex conversion framework determines the increments, $\mathbf{x}_t - \mathbf{x}_{t-1}$, based on updates from the online learner.

Practically, the online-to-nonconvex conversion framework offers a promising perspective on the success of popular optimizers. By choosing standard online learners within this framework, we recover practical algorithms: online mirror descent corresponds to SGD with momentum (Zhang and Cutkosky, 2024), while a variant of follow-the-regularized-leader (FTRL) aligns with Adam (Ahn and Cutkosky, 2024). This natural alignment with widely used optimizers underscores the framework's potential as a powerful tool for understanding the success of practical optimization algorithms. Building on this, our work further demonstrates the framework's promise by showing that it also captures the highly effective schedule-free SGD algorithm.

Lastly, we note that the existing theory for the schedule-free method is limited to convex losses, analyzed through online-to-batch conversions (Defazio et al., 2024). Specifically, when the base optimizer is selected as an online learner, Defazio et al. (2024) demonstrate that schedule-free methods effectively transform online learning guarantees into optimization guarantees for the last iterate, similar to the results of (Cutkosky, 2019; Kavis et al., 2019). However, this convex theory does not account for the empirical success of schedule-free methods in training neural networks. Our work extends these insights by showing that schedule-free SGD is also effective for nonsmooth, nonconvex optimization.

## Impact Statement

This paper studies Schedule-Free optimizer in the nonconvex setting. This work is theoretical and we do not see any immediate potential societal consequences.

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

# A. Proof of the discounted-to-nonconvex conversion (Lemma 3.1)

We begin this proof with some notations. Recall that $\tau$ is the random index distributed over $[T]$ as:

$$\Pr(\tau = t) = p_t := \begin{cases} \frac{1-\beta^t}{T}, & \text{if } t = 1, \ldots, T-1, \\ \frac{1}{1-\beta} \cdot \frac{1-\beta^T}{T}, & \text{if } t = T. \end{cases}$$

For each $t \in [T]$, let $\mathbf{Y}_t$ be the random iterate distributed over $\{\mathbf{y}_s\}_{s=1}^t$ as:

$$\Pr(\mathbf{Y}_t = \mathbf{y}_s) = q_{t,s} := \beta^{t-s} \cdot \frac{1-\beta}{1-\beta^t} \quad \text{for } s = 1, 2, \ldots, t.$$

In particular, it follows that $\mathbb{E}_{\mathbf{Y}_t}[\mathbf{Y}_t] = \frac{1-\beta}{1-\beta^t} \cdot \sum_{s=1}^t \beta^{t-s} \mathbf{y}_s = \overline{\mathbf{y}}_t$.

In order to prove Lemma 3.1, we want to upper bound the following quantity:

$$\mathbb{E}_\tau \|\nabla F(\overline{\mathbf{y}}_\tau)\|^{[\lambda]}.$$

From the definition of a $(\lambda, \varepsilon)$-stationary point (Definition 1), it follows that

$$\mathbb{E}_\tau \|\nabla F(\overline{\mathbf{y}}_\tau)\|^{[\lambda]} \le \mathbb{E}_\tau \left[ \|\mathbb{E}_{\mathbf{Y}_\tau} \nabla F(\mathbf{Y}_\tau)\| + \lambda \mathbb{E}_{\mathbf{Y}_\tau} \|\mathbf{Y}_\tau - \overline{\mathbf{y}}_\tau\|^2 \right].$$

We will now upper bound each term individually. We begin with the first term.

**Lemma A.1.** *For $\beta \in (0, 1)$, consider the iterates generated as per Algorithm 1. Consider the following definitions:*

- *For each $t \in [T]$, let $\mathbf{u}_t := -D \frac{\sum_{s=1}^t \beta^{t-s} \nabla F(\mathbf{y}_s)}{\left\| \sum_{s=1}^t \beta^{t-s} \nabla F(\mathbf{y}_s) \right\|}$.*

- *For each $t \in [T]$, let $\mathbf{Y}_t$ be the random iterate distributed over $\{\mathbf{y}_s\}_{s=1}^t$ as:*

$$\Pr(\mathbf{Y}_t = \mathbf{y}_s) = q_{t,s} := \beta^{t-s} \cdot \frac{1-\beta}{1-\beta^t} \quad \text{for } s = 1, 2, \ldots, t.$$

*Then, the following upper bound holds:*

$$\mathbb{E}_\tau \|\mathbb{E}_{\mathbf{Y}_\tau} \nabla F(\mathbf{Y}_\tau)\| \le \frac{\beta \mathbb{E}\left[\mathsf{Regret}_T^{[\beta]}(\mathbf{u}_T)\right] + (1-\beta) \sum_{t=1}^T \mathbb{E}\left[\mathsf{Regret}_t^{[\beta]}(\mathbf{u}_t)\right]}{DT} - \frac{\mu \mathbb{E} \sum_{t=1}^T \|\boldsymbol{\delta}_t\|^2}{2DT}$$
$$+ \frac{\mathbb{E} \sum_{t=1}^T (F(\mathbf{x}_t) - F(\mathbf{w}_t))}{DT} + \frac{\sigma\beta}{T\sqrt{1-\beta}} + \sigma\sqrt{1-\beta} + \frac{\mu D}{2}.$$

*Proof.* See Section A.1. □

For the second term, we use the following upper bound.

**Lemma A.2.** *For $\beta \in (0, 1)$, consider the iterates generated as per Algorithm 1*

$$\mathbb{E}_\tau \mathbb{E}_{\mathbf{Y}_\tau} \|\mathbf{Y}_\tau - \overline{\mathbf{y}}_\tau\|^2 \le \frac{2}{T} \mathbb{E} \sum_{t=1}^T \|\boldsymbol{\delta}_t\|^2 + \frac{4\beta}{(1-\beta)^2 T} \mathbb{E} \sum_{t=1}^T \|\mathbf{x}_t - \mathbf{x}_{t-1}\|^2.$$

*Proof.* See Section A.2. □

Now let us combine above two results to finish the proof. First, by Lemma A.2, we get

$$\lambda \mathbb{E}_\tau \mathbb{E}_{\mathbf{Y}_\tau} \|\mathbf{Y}_\tau - \overline{\mathbf{y}}_\tau\|^2 \le \frac{4\lambda}{T} \left( \mathbb{E} \sum_{t=1}^T \|\boldsymbol{\delta}_t\|^2 + \frac{1}{(1-\beta)^2} \mathbb{E} \sum_{t=1}^T \|\mathbf{x}_t - \mathbf{x}_{t-1}\|^2 \right).$$

Therefore, Upon combining Lemma A.1 together with Lemma A.2, we get:

$$\mathbb{E}_\tau \|\nabla F(\bar{\mathbf{y}}_\tau)\|^{[\lambda]} \le \frac{1}{DT} \left( \beta \mathbb{E} \left[ \mathsf{Regret}_T^{[\beta]}(\mathbf{u}_T) \right] + (1-\beta) \sum_{t=1}^{T} \mathbb{E} \left[ \mathsf{Regret}_t^{[\beta]}(\mathbf{u}_t) \right] \right)$$

$$+ \frac{1}{DT} \mathbb{E} \sum_{t=1}^{T} (F(\mathbf{x}_t) - F(\mathbf{w}_t))$$

$$+ \frac{4\lambda}{T} \left( \mathbb{E} \sum_{t=1}^{T} \|\boldsymbol{\delta}_t\|^2 + \frac{1}{(1-\beta)^2} \mathbb{E} \sum_{t=1}^{T} \|\mathbf{x}_t - \mathbf{x}_{t-1}\|^2 \right) - \frac{\mu}{2DT} \mathbb{E} \sum_{t=1}^{T} \|\boldsymbol{\delta}_t\|^2$$

$$+ \frac{\mu D}{2} + \frac{\sigma}{T\sqrt{1-\beta}} + \sigma\sqrt{1-\beta}.$$

Now, using the definition of $C_{\mathbf{x}}$ (see Definition 5), it follows that

$$\frac{4\lambda}{T} \left( \mathbb{E} \sum_{t=1}^{T} \|\boldsymbol{\delta}_t\|^2 + \frac{1}{(1-\beta)^2} \mathbb{E} \sum_{t=1}^{T} \|\mathbf{x}_t - \mathbf{x}_{t-1}\|^2 \right) - \frac{\mu}{2DT} \mathbb{E} \sum_{t=1}^{T} \|\boldsymbol{\delta}_t\|^2$$

$$\le \left( 4\lambda(1 + C_{\mathbf{x}}(1-\beta)^{-2}) - \frac{\mu}{2D} \right) \cdot \frac{1}{T} \mathbb{E} \sum_{t=1}^{T} \|\boldsymbol{\delta}_t\|^2 \le 0,$$

where the last line holds since $\mu \ge 8\lambda D(1 + C_{\mathbf{x}}(1-\beta)^{-2})$. Therefore, we get the desirable upper bound.

### A.1. Proof of Lemma A.1

We begin with the following identity:

$$\sum_{n=1}^{T} \sum_{t=1}^{n} \beta^{n-t}(1-\beta)(F(\mathbf{w}_t) - F(\mathbf{x}_t)) = \sum_{t=1}^{T} \sum_{n=t}^{T} \beta^{n-t}(1-\beta)(F(\mathbf{w}_t) - F(\mathbf{x}_t))$$

$$= \sum_{t=1}^{T} (1 - \beta^{T-t+1})(F(\mathbf{w}_t) - F(\mathbf{x}_t))$$

$$= \sum_{t=1}^{T} (F(\mathbf{w}_t) - F(\mathbf{x}_t)) - \sum_{t=1}^{T} \beta^{T-t+1}(F(\mathbf{w}_t) - F(\mathbf{x}_t)).$$

After rearranging and taking expectations, it follows that

$$0 = \underbrace{(1-\beta)\mathbb{E} \sum_{n=1}^{T} \sum_{t=1}^{n} \beta^{n-t}(F(\mathbf{w}_t) - F(\mathbf{x}_t))}_{=:(\mathbf{A})} + \underbrace{\beta\mathbb{E} \sum_{t=1}^{T} \beta^{T-t}(F(\mathbf{w}_t) - F(\mathbf{x}_t))}_{=:(\mathbf{B})} + \mathbb{E} \sum_{t=1}^{T} (F(\mathbf{x}_t) - F(\mathbf{w}_t)).$$

Given this decomposition, we will carefully upper bound both **(A)** and **(B)**. This will be done based on the following result.

**Lemma A.3.** *For $\beta \in (0,1)$, consider the iterates generated as per Algorithm 1. Consider the following definitions:*

- *For each $t \in [T]$, let $\mathbf{u}_t \coloneqq -D \frac{\sum_{s=1}^{t} \beta^{t-s}\nabla F(\mathbf{y}_s)}{\left\| \sum_{s=1}^{t} \beta^{t-s}\nabla F(\mathbf{y}_s) \right\|}.$*

- *For each $t \in [T]$, let $\mathbf{Y}_t$ be the random iterate distributed over $\{\mathbf{y}_s\}_{s=1}^{t}$ as:*

$$\Pr(\mathbf{Y}_t = \mathbf{y}_s) = q_{t,s} \coloneqq \beta^{t-s} \cdot \frac{1-\beta}{1-\beta^t} \quad \text{for } s = 1, 2, \ldots, t.$$

*Then, for any $n \in [T]$, we have*

$$\mathbb{E} \sum_{t=1}^{n} \beta^{n-t}(F(\mathbf{w}_t) - F(\mathbf{x}_t)) \leq -D \frac{1 - \beta^n}{1 - \beta} \mathbb{E} \left\| \mathbb{E}_{\mathbf{Y}_n} \nabla F(\mathbf{Y}_n) \right\| + \frac{\sigma D}{\sqrt{1 - \beta}}$$

$$+ \mathbb{E}[\mathsf{Regret}_n^{[\beta]}(\mathbf{u}_n)] + \mathbb{E} \sum_{t=1}^{n} \beta^{n-t}(-\frac{\mu}{2} \left\| \boldsymbol{\delta}_t \right\|^2 + \frac{\mu}{2} D^2).$$

*Proof.* See Section A.3.1. □

By Lemma A.3, it follows that

$$(\mathbf{A}) \leq -D\mathbb{E} \sum_{n=1}^{T}(1 - \beta^n) \left\| \mathbb{E}_{\mathbf{Y}_n} \nabla F(\mathbf{Y}_n) \right\| + \sigma D T \sqrt{1 - \beta} + (1 - \beta)\mathbb{E} \sum_{n=1}^{T} \mathsf{Regret}_n^{[\beta]}(\mathbf{u}_n)$$

$$+ (1 - \beta)\mathbb{E} \sum_{n=1}^{T} \sum_{t=1}^{n} \beta^{n-t}(-\frac{\mu}{2} \left\| \boldsymbol{\delta}_t \right\|^2 + \frac{\mu}{2} D^2).$$

Note that the last term can be further simplified:

$$(1 - \beta)\mathbb{E} \sum_{n=1}^{T} \sum_{t=1}^{n} \beta^{n-t}(-\frac{\mu}{2} \left\| \boldsymbol{\delta}_t \right\|^2 + \frac{\mu}{2} D^2) = (1 - \beta)\mathbb{E} \sum_{t=1}^{T} \sum_{n=t}^{T} \beta^{n-t}(-\frac{\mu}{2} \left\| \boldsymbol{\delta}_t \right\|^2 + \frac{\mu}{2} D^2)$$

$$= \mathbb{E} \sum_{t=1}^{T}(1 - \beta^{T-t+1})(-\frac{\mu}{2} \left\| \boldsymbol{\delta}_t \right\|^2 + \frac{\mu}{2} D^2).$$

Next, again using Lemma A.3, we get

$$(\mathbf{B}) \leq -D \frac{\beta - \beta^{T+1}}{1 - \beta} \mathbb{E} \left\| \mathbb{E}_{\mathbf{Y}_T} \nabla F(\mathbf{Y}_T) \right\| + \frac{\sigma \beta D}{\sqrt{1 - \beta}}$$

$$+ \beta \mathbb{E}[\mathsf{Regret}_T^{[\beta]}(\mathbf{u}_T)] + \mathbb{E} \sum_{t=1}^{T} \beta^{T-t+1}(-\frac{\mu}{2} \left\| \boldsymbol{\delta}_t \right\|^2 + \frac{\mu}{2} D^2).$$

Combining the above two inequalities for **(A)** and **(B)**, and plugging them back to the original identity, we get the following inequality:

$$0 \leq -D \cdot \mathbb{E} \left[ \frac{\beta - \beta^{T+1}}{1 - \beta} \left\| \mathbb{E}_{\mathbf{Y}_T} \nabla F(\mathbf{Y}_T) \right\| + \sum_{t=1}^{T}(1 - \beta^t) \left\| \mathbb{E}_{\mathbf{Y}_t} \nabla F(\mathbf{Y}_t) \right\| \right]$$

$$+ \sigma D T \sqrt{1 - \beta} + (1 - \beta)\mathbb{E} \sum_{t=1}^{T} \mathsf{Regret}_t^{[\beta]}(\mathbf{u}_t) + \mathbb{E} \sum_{t=1}^{T}(1 - \beta^{T-t+1})(-\frac{\mu}{2} \left\| \boldsymbol{\delta}_t \right\|^2 + \frac{\mu}{2} D^2)$$

$$+ \frac{\sigma \beta D}{\sqrt{1 - \beta}} + \beta \mathbb{E}[\mathsf{Regret}_T^{[\beta]}(\mathbf{u}_T)] + \mathbb{E} \sum_{t=1}^{T} \beta^{T-t+1}(-\frac{\mu}{2} \left\| \boldsymbol{\delta}_t \right\|^2 + \frac{\mu}{2} D^2) + \sum_{t=1}^{T}(F(\mathbf{x}_t) - F(\mathbf{w}_t)).$$

Hence, after rearranging, we get the following inequality:

$$D \cdot \mathbb{E} \left[ \frac{\beta - \beta^{T+1}}{1 - \beta} \left\| \mathbb{E}_{\mathbf{Y}_T} \nabla F(\mathbf{Y}_T) \right\| + \sum_{t=1}^{N}(1 - \beta^t) \left\| \mathbb{E}_{\mathbf{Y}_t} \nabla F(\mathbf{Y}_t) \right\| \right]$$

$$\leq (1 - \beta)\mathbb{E} \sum_{t=1}^{T} \mathsf{Regret}_t^{[\beta]}(\mathbf{u}_t) + \beta \mathbb{E}[\mathsf{Regret}_T^{[\beta]}(\mathbf{u}_T)] + \sum_{t=1}^{T}(F(\mathbf{x}_t) - F(\mathbf{w}_t))$$

$$+ \sigma D T \sqrt{1 - \beta} + \frac{\sigma \beta D}{\sqrt{1 - \beta}} + + \mathbb{E} \sum_{t=1}^{T}(-\frac{\mu}{2} \left\| \boldsymbol{\delta}_t \right\|^2 + \frac{\mu}{2} D^2).$$

In order to simplify the left hand side, notice first that

$$\frac{\beta - \beta^{T+1}}{1 - \beta} + \sum_{t=1}^{T}(1 - \beta^t) = T + \frac{\beta - \beta^{T+1}}{1 - \beta} - \frac{\beta - \beta^{T+1}}{1 - \beta} = T.$$

Let $\tau$ be the random index among $[T]$ such that

$$\Pr(\tau = t) = \begin{cases} \frac{1-\beta^t}{T} & \text{if } t = 1, \ldots, T-1, \\ \frac{1-\beta^T + \frac{\beta(1-\beta^T)}{1-\beta}}{T} = \frac{1}{1-\beta} \cdot \frac{1-\beta^T}{T} & \text{if } t = T. \end{cases}$$

Then, it follows that

$$D \cdot \mathbb{E}\left[\frac{\beta - \beta^{T+1}}{1 - \beta} \|\mathbb{E}_{\mathbf{Y}_T} \nabla F(\mathbf{Y}_T)\| + \sum_{t=1}^{T}(1 - \beta^t) \|\mathbb{E}_{\mathbf{Y}_t} \nabla F(\mathbf{Y}_t)\|\right] = DT \cdot \mathbb{E}_\tau \|\mathbb{E}_{\mathbf{Y}_\tau} \nabla F(\mathbf{Y}_\tau)\|.$$

Plugging this back to the above inequality and dividing both side by $DT$, we get the desired inequality in Lemma 3.1.

### A.2. Proof of Lemma A.2

Similar to $\mathbf{Y}_n, \overline{\mathbf{y}}_n$, we define the random iterates for the x-sequence:

- For each $n \in [T]$, define the random iterate $\mathbf{X}_n$ distributed over $\{\mathbf{x}_n\}_{s=1}^{n}$ as:

$$\Pr(\mathbf{X}_n = \mathbf{x}_s) = q_{n,s} = \beta^{n-s} \cdot \frac{1 - \beta}{1 - \beta^n} \quad \text{for } s = 1, 2, \ldots, n.$$

- Let $\overline{\mathbf{x}}_n$ be defined as:

$$\overline{\mathbf{x}}_n := \mathbb{E}_{\mathbf{X}_n}[\mathbf{X}_n] = \frac{1 - \beta}{1 - \beta^n} \cdot \sum_{s=1}^{n} \beta^{n-s} \mathbf{x}_s.$$

Given this notation, one can prove the following result.s

**Lemma A.4** (Variance decomposition lemma)**.** *For each $n \in [T]$ the following holds:*

$$\mathbb{E}_{\mathbf{Y}_n} \|\mathbf{Y}_n - \overline{\mathbf{y}}_n\|^2 \leq 2\mathbb{E}_{\mathbf{X}_n} \|\mathbf{X}_n - \overline{\mathbf{x}}_n\|^2 + 2\sum_{s=1}^{n} q_{n,s} \|\mathbf{y}_s - \mathbf{x}_s\|^2.$$

*Proof.* See Section A.3.2. $\square$

Applying Lemma A.4 to the random index $\tau$ results in

$$\mathbb{E}_\tau \mathbb{E}_{\mathbf{Y}_\tau} \|\mathbf{Y}_\tau - \overline{\mathbf{y}}_\tau\|^2 \leq 2\mathbb{E}_\tau \mathbb{E}_{\mathbf{X}_\tau} \|\mathbf{X}_\tau - \overline{\mathbf{x}}_\tau\|^2 + 2\mathbb{E}_\tau \left[\sum_{s=1}^{\tau} q_{\tau,s} \|\mathbf{y}_s - \mathbf{x}_s\|^2\right].$$

For upper bounding the first term, we use the following result.

**Lemma A.5.** *The following holds:*

$$\mathbb{E}_\tau \mathbb{E}_{\mathbf{X}_\tau} \|\mathbf{X}_\tau - \overline{\mathbf{x}}_\tau\|^2 \leq \frac{2\beta}{(1 - \beta)^2 T} \sum_{n=1}^{T} \|\mathbf{x}_n - \mathbf{x}_{n-1}\|^2.$$

*Proof.* See Section A.3.3. $\square$

Using Lemma A.5 for the first term on the RHS, we obtain:

$$2\mathbb{E}_\tau \mathbb{E}_{\mathbf{X}_\tau} \|\mathbf{X}_\tau - \overline{\mathbf{x}}_\tau\|^2 \le \frac{4\beta}{(1-\beta)^2 T} \sum_{n=1}^{T} \|\mathbf{x}_n - \mathbf{x}_{n-1}\|^2.$$

Expanding the expectation for the second term, gives us:

$$2\mathbb{E}_\tau \left[\sum_{s=1}^{\tau} q_{\tau,s} \|\mathbf{y}_s - \mathbf{x}_s\|^2\right] = 2\sum_{n=1}^{T}\sum_{s=1}^{n} p_n q_{n,s} \|\mathbf{y}_s - \mathbf{x}_s\|^2$$

$$= 2\sum_{s=1}^{T} \left(\sum_{n=s}^{T} p_n q_{n,s}\right) \|\mathbf{y}_s - \mathbf{x}_s\|^2.$$

Finally, plugging in the values of $p_n$ and $q_{n,s}$, we get:

$$\sum_{n=s}^{T} p_n q_{n,s} = \sum_{n=s}^{T-1} (p_n q_{n,s}) + p_T q_{T,s}$$

$$= \sum_{n=s}^{T-1} \left(\frac{1-\beta^n}{T} \cdot \left(\beta^{n-s}\frac{1-\beta}{1-\beta^n}\right)\right) + \frac{1}{1-\beta}\frac{1-\beta^T}{T} \cdot \left(\beta^{T-s} \cdot \frac{1-\beta}{1-\beta^T}\right)$$

$$= \frac{1}{T}\sum_{n=s}^{T-1} \beta^{n-s}(1-\beta) + \frac{\beta^{T-s}}{T}$$

$$= \frac{1-\beta^{T-s}}{T} + \frac{\beta^{T-s}}{T} = \frac{1}{T}.$$

Putting these two upper bounds together and using the fact that $\mathbb{E} \|\mathbf{w}_s - \mathbf{x}_s\|^2 = \mathbb{E} \|\boldsymbol{\delta}_s\|^2$, our final bound follows.

### A.3. Proofs of technical lemmas

#### A.3.1. PROOF OF LEMMA A.3

For an index $t$, let us first consider the term $\mathbb{E}[F(\mathbf{w}_t) - F(\mathbf{x}_t)]$. By 1 and the fact that $\mathbf{w}_t - \mathbf{x}_t = \boldsymbol{\delta}_t$, together with the fact that $\mathbf{y}_t = \mathbf{x}_t + s_t\boldsymbol{\delta}_t$ for $s_t \sim \text{Unif}[0, 1]$ (see Algorithm 1), we have the following:

$$\mathbb{E}_{s_t}[F(\mathbf{w}_t) - F(\mathbf{x}_t)] = \mathbb{E}_{s_t}\langle \nabla F(\mathbf{y}_t), \boldsymbol{\delta}_t\rangle.$$

On the other hand, for $n \ge t$, it holds under the randomness over stochastic gradient $\mathbf{g}_t$ that

$$\mathbb{E}_{\mathbf{g}_t}\langle \nabla F(\mathbf{y}_t), \boldsymbol{\delta}_t\rangle = \mathbb{E}_{\mathbf{g}_t}\langle \nabla F(\mathbf{y}_t), \mathbf{u}_n\rangle + \mathbb{E}_{\mathbf{g}_t}\langle \nabla F(\mathbf{y}_t), \boldsymbol{\delta}_t - \mathbf{u}_n\rangle$$

$$= \mathbb{E}_{\mathbf{g}_t}\langle \nabla F(\mathbf{y}_t), \mathbf{u}_n\rangle + \mathbb{E}_{\mathbf{g}_t}\langle \nabla F(\mathbf{y}_t) - \mathbf{g}_t, \boldsymbol{\delta}_t - \mathbf{u}_n\rangle + \langle \mathbf{g}_t, \boldsymbol{\delta}_t - \mathbf{u}_n\rangle$$

$$= \mathbb{E}_{\mathbf{g}_t}\langle \nabla F(\mathbf{y}_t), \mathbf{u}_n\rangle + \mathbb{E}_{\mathbf{g}_t}\langle \nabla F(\mathbf{y}_t) - \mathbf{g}_t, -\mathbf{u}_n\rangle + \mathbb{E}_{\mathbf{g}_t}\langle \mathbf{g}_t, \boldsymbol{\delta}_t - \mathbf{u}_n\rangle,$$

where the last line follows since

$$\mathbb{E}_{\mathbf{g}_t}\langle \nabla F(\mathbf{y}_t) - \mathbf{g}_t, \boldsymbol{\delta}_t\rangle = \langle \mathbb{E}_{\mathbf{g}_t}[\nabla F(\mathbf{y}_t) - \mathbf{g}_t], \boldsymbol{\delta}_t\rangle = 0.$$

Upon taking the overall expectation, we have

$$\mathbb{E}[F(\mathbf{w}_t) - F(\mathbf{x}_t)] = \mathbb{E}\left[\langle \nabla F(\mathbf{y}_t), \mathbf{u}_n\rangle + \langle \nabla F(\mathbf{y}_t) - \mathbf{g}_t, -\mathbf{u}_n\rangle + \langle \mathbf{g}_t, \boldsymbol{\delta}_t - \mathbf{u}_n\rangle\right].$$

Hence, it follows that

$$\mathbb{E}\sum_{t=1}^{n} \beta^{n-t}(F(\mathbf{w}_t) - F(\mathbf{x}_t)) = \mathbb{E}\sum_{t=1}^{n} \beta^{n-t}\left[\langle \nabla F(\mathbf{y}_t), \mathbf{u}_n\rangle + \langle \nabla F(\mathbf{y}_t) - \mathbf{g}_t, -\mathbf{u}_n\rangle + \langle \mathbf{g}_t, \boldsymbol{\delta}_t - \mathbf{u}_n\rangle\right]. \tag{9}$$

Let us consider each term on the right hand side one by one:

- Using the definition $\mathbf{u}_n := -D\frac{\sum_{t=1}^{n}\beta^{n-t}\nabla F(\mathbf{y}_t)}{\|\sum_{t=1}^{n}\beta^{n-t}\nabla F(\mathbf{y}_t)\|}$, the first term can be expressed as:

$$\mathbb{E}\sum_{t=1}^{n}\beta^{n-t}\langle\nabla F(\mathbf{y}_t),\mathbf{u}_n\rangle = \mathbb{E}\left\langle\sum_{t=1}^{n}\beta^{n-t}\nabla F(\mathbf{y}_t),-D\frac{\sum_{t=1}^{n}\beta^{n-t}\nabla F(\mathbf{y}_t)}{\|\sum_{t=1}^{n}\beta^{n-t}\nabla F(\mathbf{y}_t)\|}\right\rangle \tag{10}$$

$$= -D\mathbb{E}\left\|\sum_{t=1}^{n}\beta^{n-t}\nabla F(\mathbf{y}_t)\right\| = -D\frac{1-\beta^n}{1-\beta}\mathbb{E}\left\|\mathbb{E}_{\mathbf{Y}_n}\nabla F(\mathbf{Y}_n)\right\|. \tag{11}$$

- For the second term, using Cauchy-Schwartz inequality, we have

$$\mathbb{E}\sum_{t=1}^{n}\beta^{n-t}\langle\nabla F(\mathbf{y}_t)-\mathbf{g}_t,-\mathbf{u}_n\rangle \leq \sqrt{\mathbb{E}\left\|\sum_{t=1}^{n}\beta^{n-t}(\nabla F(\mathbf{y}_t)-\mathbf{g}_t)\right\|^2\mathbb{E}\|\mathbf{u}_n\|^2}. \tag{12}$$

Using the bounded variance assumption on the stochastic gradient oracle, we have

$$\mathbb{E}\left\|\sum_{t=1}^{n}\beta^{n-t}(\nabla F(\mathbf{y}_t)-\mathbf{g}_t)\right\|^2 = \mathbb{E}\sum_{t=1}^{n}\beta^{2(n-t)}\|\nabla F(\mathbf{y}_t)-\mathbf{g}_t\|^2 \leq \frac{\sigma^2}{1-\beta^2} \leq \frac{\sigma^2}{1-\beta}. \tag{13}$$

where we used the fact $\frac{1}{1-\beta^2} \leq \frac{1}{1-\beta}$. Thus, it holds that

$$\mathbb{E}\sum_{t=1}^{n}\beta^{n-t}\langle\nabla F(\mathbf{y}_t)-\mathbf{g}_t,-\mathbf{u}_n\rangle \leq \frac{\sigma D}{\sqrt{1-\beta}}.$$

- For the third term, we have

$$\mathbb{E}\sum_{t=1}^{n}\beta^{n-t}\langle\mathbf{g}_t,\boldsymbol{\delta}_t-\mathbf{u}_n\rangle = \mathbb{E}[\mathsf{Regret}_n^{[\beta]}(\mathbf{u}_n)] + \mathbb{E}\sum_{t=1}^{n}\beta^{n-t}(-\frac{\mu}{2}\|\boldsymbol{\delta}_t\|^2 + \frac{\mu}{2}\|\mathbf{u}_n\|^2)$$

$$= \mathbb{E}[\mathsf{Regret}_n^{[\beta]}(\mathbf{u}_n)] + \mathbb{E}\sum_{t=1}^{n}\beta^{n-t}(-\frac{\mu}{2}\|\boldsymbol{\delta}_t\|^2 + \frac{\mu}{2}D^2).$$

This completes the proof of Lemma A.3.

### A.3.2. PROOF OF LEMMA A.4

Let $S_n$ be a random index distributed over $1, 2, \ldots, n$ such that $\Pr(S_n = s) = q_{n,s}$. Moreover, let $\widehat{S}_n$ be an independent copy of $S_n$. Then, it follows that

$$\mathbb{E}_{\mathbf{Y}_n}\|\mathbf{Y}_n-\overline{\mathbf{y}}_n\|^2 = \mathbb{E}_{S_n}\left\|\mathbf{y}_{S_n}-\mathbb{E}_{\widehat{S}_n}[\mathbf{y}_{\widehat{S}_n}]\right\|^2$$

$$= \mathbb{E}_{S_n}\left\|\mathbf{x}_{S_n}-\mathbb{E}_{\widehat{S}_n}[\mathbf{x}_{\widehat{S}_n}]+\mathbf{y}_{S_n}-\mathbf{x}_{S_n}-\mathbb{E}_{\widehat{S}_n}[\mathbf{y}_{\widehat{S}_n}-\mathbf{x}_{\widehat{S}_n}]\right\|^2$$

$$\leq 2\mathbb{E}_{S_n}\left\|\mathbf{x}_{S_n}-\mathbb{E}_{\widehat{S}_n}[\mathbf{x}_{\widehat{S}_n}]\right\|^2 + 2\mathbb{E}_{S_n}\left\|\mathbf{y}_{S_n}-\mathbf{x}_{S_n}-\mathbb{E}_{\widehat{S}_n}[\mathbf{y}_{\widehat{S}_n}-\mathbf{x}_{\widehat{S}_n}]\right\|^2$$

$$\leq 2\mathbb{E}_{S_n}\left\|\mathbf{x}_{S_n}-\mathbb{E}_{\widehat{S}_n}[\mathbf{x}_{\widehat{S}_n}]\right\|^2 + 2\mathbb{E}_{S_n}\|\mathbf{y}_{S_n}-\mathbf{x}_{S_n}\|^2$$

$$= 2\mathbb{E}_{\mathbf{X}_n}\|\mathbf{X}_n-\overline{\mathbf{x}}_n\|^2 + 2\sum_{s=1}^{n}q_{n,s}\|\mathbf{y}_s-\mathbf{x}_s\|^2.$$

where for the first bound we have used $\|x+y\|^2 \leq 2\|x\|^2 + 2\|y\|^2$, and for the second one we applied $\mathbb{E}\|\mathbf{X}-\mathbb{E}\mathbf{X}\|^2 = \mathbb{E}\|\mathbf{X}\|^2 - \|\mathbb{E}\mathbf{X}\|^2 \leq \mathbb{E}\|\mathbf{X}\|^2$. This completes the proof.

### A.3.3. PROOF OF LEMMA A.5

We start with the following two results.

**Proposition A.1.** *For each $n \in [T]$, it holds that*

$$\mathbb{E}_{\mathbf{X}_n} \|\mathbf{X}_n - \overline{\mathbf{x}}_n\|^2 \le 2 \sum_{t=1}^{n} \lambda_{n,t} \|\mathbf{x}_t - \mathbf{x}_{t-1}\|^2, \quad \text{where } \lambda_{n,t} := \sum_{i=t}^{n} \sum_{j=1}^{t-1} q_{n,i} q_{n,j}(i-j).$$

**Proof of Proposition A.1.** Let us fix $n \in [T]$. Let $\widehat{\mathbf{X}}_n$ be an independent copy of $\mathbf{X}_n$. Then, by Jensen's inequality, we get:

$$\mathbb{E}_{\mathbf{X}_n} \|\mathbf{X}_n - \overline{\mathbf{x}}_n\|^2 = \mathbb{E}_{\mathbf{X}_n} \left\| \mathbf{X}_n - \mathbb{E}_{\widehat{\mathbf{X}}_n}[\widehat{\mathbf{X}}_n] \right\|^2 \le \mathbb{E}_{\mathbf{X}_n, \widehat{\mathbf{X}}_n} \left\| \mathbf{X}_n - \widehat{\mathbf{X}}_n \right\|^2 \tag{14}$$

$$= 2 \sum_{i=1}^{n} \sum_{j=1}^{i-1} q_{n,i} q_{n,j} \|\mathbf{x}_i - \mathbf{x}_j\|^2. \tag{15}$$

Next, using Cauchy-Schwartz and triangle inequality, we get:

$$\|\mathbf{x}_i - \mathbf{x}_j\|^2 \le \left( \sum_{t=j+1}^{i} \|\mathbf{x}_t - \mathbf{x}_{t-1}\| \right)^2 \le (i-j) \sum_{t=j+1}^{i} \|\mathbf{x}_t - \mathbf{x}_{t-1}\|^2.$$

Hence, it follows that

$$\mathbb{E}_{\mathbf{X}_n} \|\mathbf{X}_n - \overline{\mathbf{x}}_n\|^2 \le 2 \sum_{i=1}^{n} \sum_{j=1}^{i-1} \sum_{t=j+1}^{i} q_{n,i} q_{n,j}(i-j) \|\mathbf{x}_t - \mathbf{x}_{t-1}\|^2$$

$$= 2 \sum_{t=1}^{n} \left( \sum_{i=t}^{n} \sum_{j=1}^{t-1} q_{n,i} q_{n,j}(i-j) \right) \|\mathbf{x}_t - \mathbf{x}_{t-1}\|^2$$

This completes the proof. $\square$

**Proposition A.2.** *For $1 \le t \le n \le T$, it holds that*

$$\lambda_{n,t} \le \frac{(n-t+1)\beta^{n-t+1}}{1 - \beta^n}.$$

**Proof of Proposition A.2.** In order to prove the desired upper bound, let us first simplify the expression for $\lambda_{n,t}$ as follows:

$$\lambda_{n,t} = \left( \frac{1-\beta}{1-\beta^n} \right)^2 \cdot \sum_{i=t}^{n} \sum_{j=1}^{t-1} \beta^{n-i} \beta^{n-j}(i-j)$$

$$= \left( \frac{1-\beta}{1-\beta^n} \right)^2 \cdot \sum_{i=t}^{n} \sum_{\delta=i-t+1}^{i-1} \beta^{n-i} \beta^{n-i+\delta} \delta \qquad (\delta := i-j)$$

$$= \left( \frac{1-\beta}{1-\beta^n} \right)^2 \cdot \sum_{k=0}^{n-t} \sum_{\delta=n-k-t+1}^{n-k-1} \beta^{k} \beta^{k+\delta} \delta \qquad (k := n-i)$$

$$= \left( \frac{1-\beta}{1-\beta^n} \right)^2 \cdot \sum_{k=0}^{n-t} \left[ \beta^{2k} \sum_{\delta=n-k-t+1}^{n-k-1} \delta \beta^{\delta} \right].$$

Next, let us compute the inner summation:

$$\sum_{\delta=n-k-t+1}^{n-k-1} \delta \beta^{\delta} = \beta \cdot \frac{\mathrm{d}}{\mathrm{d}\beta} \sum_{\delta=n-k-t+1}^{n-k-1} \beta^{\delta} = \beta \cdot \frac{\mathrm{d}}{\mathrm{d}\beta} \left( \frac{\beta^{n-k-t+1} - \beta^{n-k}}{1-\beta} \right)$$

$$= \frac{\beta^{a-k+1} - \beta^{b-k+1}}{(1-\beta)^2} + \frac{(a-k)\beta^{a-k} - (b-k)\beta^{b-k}}{1-\beta}. \qquad (a := n-t+1, \ b := n)$$

Substituting this back to the above expression for $\lambda_{n,t}$, we have

$$
\lambda_{n,t} = \left(\frac{1-\beta}{1-\beta^n}\right)^2 \cdot \sum_{k=0}^{n-t} \beta^{2k} \left[\frac{\beta^{a-k+1} - \beta^{b-k+1}}{(1-\beta)^2} + \frac{(a-k)\beta^{a-k} - (b-k)\beta^{b-k}}{1-\beta}\right]
$$

$$
= \left(\frac{1-\beta}{1-\beta^n}\right)^2 \cdot \sum_{k=0}^{n-t} \left[\left(\frac{\beta^{a+1} - \beta^{b+1}}{(1-\beta)^2} + \frac{a\beta^a - b\beta^b}{1-\beta}\right)\beta^k - \frac{\beta^a - \beta^b}{1-\beta} k\beta^k\right].
$$

Now one can compute the first and second terms above using the following identities:

- First term: note that $\sum_{k=0}^{n-t} \beta^k = \frac{1-\beta^{n-t+1}}{1-\beta} = \frac{1-\beta^a}{1-\beta}$.

- Second term: note that $\sum_{k=0}^{n-t} k\beta^k = \beta\frac{\mathrm{d}}{\mathrm{d}\beta}\left(\frac{1-\beta^a}{1-\beta}\right) = \frac{\beta - \beta^{a+1}}{(1-\beta)^2} - \frac{a\beta^a}{1-\beta}$.

Thus, it follows that

$$
\lambda_{n,t} = \left(\frac{1-\beta}{1-\beta^n}\right)^2 \cdot \left[\left(\frac{\beta^{a+1} - \beta^{b+1}}{(1-\beta)^2} + \frac{a\beta^a - b\beta^b}{1-\beta}\right)\cdot\frac{1-\beta^a}{1-\beta} - \frac{\beta^a - \beta^b}{1-\beta}\cdot\left(\frac{\beta - \beta^{a+1}}{(1-\beta)^2} - \frac{a\beta^a}{1-\beta}\right)\right]
$$

$$
= \frac{(\beta^{a+1} - \beta^{b+1})\frac{1-\beta^a}{1-\beta} + (a\beta^a - b\beta^b)(1-\beta^a) - (\beta^{a+1} - \beta^{b+1})\frac{1-\beta^a}{1-\beta} + (\beta^a - \beta^b)a\beta^a}{(1-\beta^n)^2}
$$

$$
= \frac{(a\beta^a - b\beta^b)(1-\beta^a) + (\beta^a - \beta^b)a\beta^a}{(1-\beta^n)^2} = \frac{a\beta^a(1-\beta^b) - b\beta^b(1-\beta^a)}{(1-\beta^n)^2} \le \frac{a\beta^a(1-\beta^b)}{(1-\beta^n)^2}.
$$

Upon plugging back the definition of $a$ and $b$, this completes the proof of Proposition A.2. $\qquad\square$

For the remainder of the proof, let us use the following notation for simplicity:

$$
\delta_t := \|\mathbf{x}_t - \mathbf{x}_{t-1}\|. \tag{16}
$$

Combining Proposition A.1 and Proposition A.2, it follows that

$$
\mathbb{E}_{\mathbf{X}_n} \|\mathbf{X}_n - \overline{\mathbf{x}}_n\|^2 \le 2\sum_{t=1}^{n} \frac{(n-t+1)\beta^{n-t+1}}{1-\beta^n}\delta_t^2. \tag{17}
$$

With this bound, one can upper bound the overall averaged variance term as follows:

$$
\mathbb{E}_\tau \mathbb{E}_{\mathbf{X}_\tau} \|\mathbf{X}_\tau - \overline{\mathbf{x}}_\tau\|^2 = \sum_{n=1}^{T-1}\left[p_n \mathbb{E}_{\mathbf{X}_n}\|\mathbf{X}_n - \overline{\mathbf{x}}_n\|^2\right] + p_T \mathbb{E}_{\mathbf{X}_T}\|\mathbf{X}_T - \overline{\mathbf{x}}_T\|^2 \tag{18}
$$

$$
\le 2\sum_{n=1}^{T-1}\left[\frac{1-\beta^n}{T}\sum_{t=1}^{n}\frac{(n-t+1)\beta^{n-t+1}}{1-\beta^n}\delta_t^2\right] + \frac{2(1-\beta^T)}{(1-\beta)T}\sum_{n=1}^{T}\frac{(T-n+1)\beta^{T-n+1}}{1-\beta^T}\delta_n^2 \tag{19}
$$

$$
\le \frac{2}{T}\sum_{n=1}^{T-1}\sum_{t=1}^{n}\left[(n-t+1)\beta^{n-t+1}\delta_t^2\right] + \frac{2}{(1-\beta)T}\sum_{n=1}^{T}(T-n+1)\beta^{T-n+1}\delta_n^2. \tag{20}
$$

For the first term above, note that

$$
\begin{aligned}
\sum_{n=1}^{T-1} \sum_{t=1}^{n} \left[ (n-t+1)\beta^{n-t+1}\delta_t^2 \right] &= \sum_{t=1}^{T-1} \left[ \delta_t^2 \sum_{n=t}^{T-1} (n-t+1)\beta^{n-t+1} \right] \\
&= \sum_{t=1}^{T-1} \left[ \delta_t^2 \sum_{k=1}^{T-t} k\beta^k \right] \\
&= \sum_{t=1}^{T-1} \left[ \delta_t^2 \beta \frac{\mathrm{d}}{\mathrm{d}\beta} \left( \sum_{k=1}^{T-t} \beta^k \right) \right] = \sum_{t=1}^{T-1} \left[ \delta_t^2 \beta \frac{\mathrm{d}}{\mathrm{d}\beta} \left( \frac{\beta - \beta^{T-t+1}}{1-\beta} \right) \right] \\
&= \sum_{t=1}^{T-1} \left[ \delta_t^2 \left( \frac{\beta^2 - \beta^{T-t+2}}{(1-\beta)^2} + \frac{\beta - (T-t+1)\beta^{T-t+1}}{1-\beta} \right) \right] \\
&= \sum_{n=1}^{T-1} \left( \frac{\beta^2(1-\beta^{T-n})}{(1-\beta)^2} + \frac{\beta}{1-\beta} - \frac{(T-n+1)\beta^{T-n+1}}{1-\beta} \right) \delta_n^2.
\end{aligned}
$$

Upon plugging this back to the upper bound on the overall averaged variance, we get:

$$
\begin{aligned}
\mathbb{E}_\tau \mathbb{E}_{\mathbf{X}_\tau} \|\mathbf{X}_\tau - \bar{\mathbf{x}}_\tau\|^2 &\leq \frac{2}{T} \sum_{n=1}^{T-1} \left[ \left( \frac{\beta^2(1-\beta^{T-n})}{(1-\beta)^2} + \frac{\beta}{1-\beta} \right) \delta_n^2 \right] + \frac{2\beta}{(1-\beta)T} \delta_T^2 \\
&\leq \frac{2}{T} \sum_{n=1}^{T-1} \left[ \left( \frac{\beta^2 + \beta(1-\beta)}{(1-\beta)^2} \right) \delta_n^2 \right] + \frac{2\beta}{(1-\beta)T} \delta_T^2 \\
&\leq \frac{2\beta}{(1-\beta)^2 T} \sum_{n=1}^{T} \delta_n^2.
\end{aligned}
$$

This completes the proof of Lemma A.5.

## B. Proof of the user-friendly nonconvex guarantees (Theorem 3.1)

Let us first recall the upper bound from Lemma 3.1:

$$
\begin{aligned}
\mathbb{E}_\tau \left[ \|\nabla F(\bar{\mathbf{y}}_\tau)\|^{[\lambda]} \right] &\leq \frac{1}{DT} \left( \beta \mathbb{E} \left[ \mathsf{Regret}_T^{[\beta]}(\mathbf{u}_T) \right] + (1-\beta) \sum_{t=1}^{T} \mathbb{E} \left[ \mathsf{Regret}_t^{[\beta]}(\mathbf{u}_t) \right] \right) \\
&\quad + \frac{1}{DT} \mathbb{E} \left[ \sum_{t=1}^{T} (F(\mathbf{x}_t) - F(\mathbf{w}_t)) \right] + \frac{\mu D}{2} + \frac{\sigma}{T\sqrt{1-\beta}} + \sigma\sqrt{1-\beta}.
\end{aligned}
$$

Below, we further upper bound the discounted regret terms. First, since $\|\mathbf{u}_n\| = D$, Lemma 3.2 implies:

$$
\forall t \in [T], \quad \beta \mathbb{E} \left[ \mathsf{Regret}_t^{[\beta]}(\mathbf{u}_t) \right] \leq \frac{2D(G+\sigma)}{\sqrt{1-\beta}} + \frac{\mu}{2}D^2, \tag{21}
$$

where this upper bound holds for all $t \in [T]$ since the upper bound does not depend on $T$.

To simplify the upper bound on the regret term, we impose the following condition:

$$
\frac{1}{2} \leq \beta \leq 1 - \frac{1}{T}. \tag{22}
$$

Additionally, we introduce the following notation to streamline our expressions:

$$
\boxed{\alpha := 1 - \beta.} \tag{23}
$$

In the sequel, we will select $1 - \beta = O(\varepsilon^2)$, so readers can treat $\alpha$ as a small quantity.

With the condition (22) together with the simplifying notation, we have $\beta^{-1} \leq 2$ and $\alpha T \geq 1$. Therefore, plugging the upper bound (21) into the discounted regret term, it follows that:

$$\frac{1}{DT} \left( \beta \mathbb{E} \left[ \mathsf{Regret}_T^{[\beta]}(\mathbf{u}_T) \right] + \alpha \sum_{t=1}^{T} \mathbb{E} \left[ \mathsf{Regret}_t^{[\beta]}(\mathbf{u}_t) \right] \right) \leq \frac{(2\alpha T + 1)}{DT} \left( \frac{2D(G + \sigma)}{\sqrt{\alpha}} + \frac{1}{2}\mu D^2 \right)$$

$$\leq \frac{3\alpha T}{DT} \left( \frac{2D(G + \sigma)}{\sqrt{\alpha}} + \frac{1}{2}\mu D^2 \right) = 6(G + \sigma)\sqrt{\alpha} + \frac{3}{2}\alpha\mu D \leq 6(G + \sigma)\sqrt{\alpha} + \frac{3}{2}\mu D.$$

Now let us plug this bound back to Lemma 3.1. For the regularization strength, we choose

$$\mu = \mu_\star := 8\lambda D \left( 1 + \sqrt{C_{\mathbf{x}}}\alpha^{-1} \right)^2.$$

This is a valid choice for Lemma 3.1 since $8\lambda D \left( 1 + \sqrt{C_{\mathbf{x}}}\alpha^{-1} \right)^2 \geq 8\lambda D \left( 1 + C_{\mathbf{x}}\alpha^{-2} \right)$. Thus, we arrive at the following bound:

$$\mathbb{E}_\tau \left[ \|\nabla F(\overline{\mathbf{y}}_\tau)\|^{[\lambda]} \right] \leq \frac{\mathbb{E} \left[ \sum_{t=1}^{T}(F(\mathbf{x}_t) - F(\mathbf{w}_t)) \right]}{DT} + \underbrace{\frac{\sigma}{T\sqrt{\alpha}}}_{\textbf{(A)}} + \underbrace{7(G + \sigma)\sqrt{\alpha}}_{\textbf{(B)}} + \underbrace{16\lambda D^2(1 + \sqrt{C_{\mathbf{x}}}\alpha^{-1})^2}_{\textbf{(C)}}.$$

Now, we show that the choice of paramters in Theorem 3.1 (*i.e.*, $\beta_\star, D_\star$ defined in the statement) leads to each of the underbraced terms is bounded by $\varepsilon$:

- **(B):** Choosing $\beta = \beta_\star = 1 - \left( \frac{\varepsilon}{7(G+\sigma)} \right)^2$, we have $\alpha = \left( \frac{\varepsilon}{7(G+\sigma)} \right)^2$. Hence, term **(B)** is at most $\varepsilon$.

- **(C):** Taking $D = D_\star = \frac{1}{4}\lambda^{-1/2}\varepsilon^{1/2}(1 + \sqrt{C_{\mathbf{x}}}\alpha^{-1})^{-1}$, we have

$$\textbf{(C)} = 16\lambda(1 + \sqrt{C_{\mathbf{x}}}\alpha^{-1})^2 D_\star^2 = \varepsilon.$$

Hence, term **(C)** is at most $\varepsilon$. Expanding with $\alpha = \left( \frac{\varepsilon}{7(G+\sigma)} \right)^2$, we have:

$$D = D_\star = \frac{1}{4}\lambda^{-1/2}\varepsilon^{1/2} \left( 1 + \frac{49(G + \sigma)^2}{\varepsilon^2}\sqrt{C_{\mathbf{x}}} \right)^{-1}.$$

- **(A):** Taking $T \geq \sigma\alpha^{-1/2}\varepsilon^{-1}$, term **(A)** is at most $\varepsilon$.

Therefore, we have the following inequality as desired:

$$\mathbb{E}_\tau \left[ \|\nabla F(\overline{\mathbf{y}}_\tau)\|^{[\lambda]} \right] \leq 3\varepsilon + \frac{\mathbb{E} \left[ \sum_{t=1}^{T}(F(\mathbf{x}_t) - F(\mathbf{w}_t)) \right]}{D_\star T}.$$

Finally, let us reconcile these parameter choices with the earlier condition (22) on $\beta$. As long as $\varepsilon \leq \frac{7}{2}(G + \sigma)$, it holds that $\alpha \leq \frac{1}{4}$, implying $\beta \geq \frac{3}{4}$, which satisfies the condition $\beta \geq \frac{1}{2}$. Additionally, to ensure $\beta \leq 1 - \frac{1}{T}$, we impose $T \geq \alpha^{-1}$. Therefore, the overall condition on $T$ becomes:

$$T \geq \max \left\{ \sigma\alpha^{-1/2}\varepsilon^{-1}, \ \alpha^{-1} \right\} = \max \left\{ 7\sigma(G + \sigma)\varepsilon^{-2}, \ 49(G + \sigma)^2\varepsilon^{-2} \right\} = 49(G + \sigma)^2\varepsilon^{-2}.$$

This completes the proof of Theorem 3.1.

## C. Proof of Corollary 5.1

We first recall the choice of parameters from Theorem 3.1 for the reader's convenience.

- The discount factor is $\beta_\star = 1 - \left(\frac{\varepsilon}{7(G+\sigma)}\right)^2$.
- The comparator norm is $D_\star = \frac{1}{4}\lambda^{-1/2}\varepsilon^{1/2}\left(1 + \frac{49(G+\sigma)^2}{\varepsilon^2}\sqrt{C_\mathbf{x}}\right)^{-1}$.
- The regularization strength is $\mu_\star = 2\lambda^{1/2}\varepsilon^{1/2}\left(1 + \frac{49(G+\sigma)^2}{\varepsilon^2}\sqrt{C_\mathbf{x}}\right)$.
- The step size of $\beta$-OMD is $\eta_\star = \frac{2}{G+\sigma}D_\star\sqrt{1-\beta_\star}$.

We now show that $C_\mathbf{x}$ is bounded by 4. First, note that with the above choice of parameters, we have

$$\zeta = \zeta_\star := \frac{\beta_\star}{1 + \eta_\star\mu_\star}. \tag{24}$$

Since $\varepsilon \le \frac{7}{2}(G+\sigma)$, it follows that $\beta_\star \ge 1 - \frac{1}{4} = \frac{3}{4}$. Moreover, we have

$$\eta_\star\mu_\star = \frac{2}{G+\sigma}D_\star\sqrt{1-\beta_\star} \cdot 2\lambda^{1/2}\varepsilon^{1/2}\left(1 + \frac{49(G+\sigma)^2}{\varepsilon^2}\sqrt{C_\mathbf{x}}\right) \tag{25}$$

$$= \frac{2}{G+\sigma} \cdot \frac{1}{4}\lambda^{-1/2}\varepsilon^{1/2} \cdot \left(\frac{\varepsilon}{7(G+\sigma)}\right) \cdot 2\lambda^{1/2}\varepsilon^{1/2} \tag{26}$$

$$= \frac{\varepsilon^2}{7(G+\sigma)^2} \le \frac{7}{4}. \tag{27}$$

This shows that

$$\zeta_\star^{-1} = \frac{1 + \eta_\star\mu_\star}{\beta_\star} \le \frac{11/4}{3/4} \le 4. \tag{28}$$

Thus, from Algorithm 5, we have $\|\mathbf{x}_t - \mathbf{x}_{t-1}\| = \|\zeta_\star^{-1}\boldsymbol{\delta}_t\|$ for $t \ge 1$, leading to the following inequality:

$$\mathbb{E}\sum_{t=1}^T \|\mathbf{x}_t - \mathbf{x}_{t-1}\|^2 = \zeta_\star^{-2}\mathbb{E}\sum_{t=1}^T \|\boldsymbol{\delta}_t\|^2 \le 16\mathbb{E}\sum_{t=1}^T \|\boldsymbol{\delta}_t\|^2, \tag{29}$$

which shows that $C_\mathbf{x} \le 16$.

Next, we upper bound the sum of loss decrements. We begin with the following decomposition, using the $G$-Lipschitzness of $F$:

$$\sum_{t=1}^T (F(\mathbf{x}_t) - F(\mathbf{w}_t)) = \sum_{t=1}^T (F(\mathbf{x}_t) - F(\mathbf{w}_{t-1}) + F(\mathbf{w}_{t-1}) - F(\mathbf{w}_t)) \tag{30}$$

$$\le \sum_{t=1}^T G\|\mathbf{x}_t - \mathbf{w}_{t-1}\| + \sum_{t=1}^T (F(\mathbf{w}_{t-1}) - F(\mathbf{w}_t)). \tag{31}$$

Thus, using the closeness property (7) between $\mathbf{x}_t$ and $\mathbf{w}_{t-1}$, it follows that

$$\mathbb{E}\sum_{t=1}^T (F(\mathbf{x}_t) - F(\mathbf{w}_t)) \le G\mathbb{E}\sum_{t=2}^T \eta_\star\|\mathbf{g}_{t-1}\| + \mathbb{E}\left[F(\mathbf{w}_0) - F(\mathbf{w}_T)\right] \tag{32}$$

$$\le G\eta_\star T(G+\sigma) + \Delta_F = 2GD_\star T\sqrt{1-\beta_\star} + \Delta_F. \tag{33}$$

Finally, let us substitute this upper bound, together with $C_{\mathbf{x}} \leq 16$, back into the inequality in Theorem 3.1. We obtain

$$
\begin{aligned}
\mathbb{E}_\tau \left[ \|\nabla F(\overline{\mathbf{y}}_\tau)\|^{[\lambda]} \right] &= 3\varepsilon + \frac{1}{D_\star T} \mathbb{E} \sum_{t=1}^{T} \left( F(\mathbf{x}_t) - F(\mathbf{w}_t) \right) \\
&\leq 3\varepsilon + \frac{1}{D_\star T} \left( 2 G D_\star T \sqrt{1 - \beta_\star} + \Delta_F \right) \\
&\leq 3\varepsilon + 2G \frac{\varepsilon}{7(G+\sigma)} + \frac{4\lambda^{1/2} \varepsilon^{-1/2} \left( 1 + \frac{49(G+\sigma)^2}{\varepsilon^2} \cdot \sqrt{C_{\mathbf{x}}} \right)}{T} \cdot \Delta_F \\
&\leq 4\varepsilon + \frac{4\Delta_F \lambda^{1/2} \varepsilon^{-1/2} \left( 5 \cdot \frac{49(G+\sigma)^2}{\varepsilon^2} \right)}{T},
\end{aligned}
$$

provided that $T \geq 49(G+\sigma)^2 \varepsilon^{-2}$.

Thus, the desired iteration complexity bound follows.

# D. Deferred proofs of other results

## D.1. Proof of the OMD discounted regret (Lemma 3.2)

To begin, let $\mathbf{v}_1, \mathbf{v}_2, \ldots, \mathbf{v}_T$ be a sequence of vectors, and consider the quadratically regularized linear loss sequence $\{\ell_t^{\mathbf{v}}\}_{t=1}^{T}$, defined as:

$$
\forall t \in [T], \quad \ell_t^{\mathbf{v}}(\cdot) = \langle \mathbf{v}_t, \cdot \rangle + \frac{\mu_t}{2} \|\cdot\|^2,
$$

where the regularization strengths $\mu_t \geq 0$ for all $t \in [T]$ are given and known to the online learner. Since the quadratic regularization term is fully known, it is standard for the online learner to incur regret only from the uncertainty in $\mathbf{v}_t$.

For instance, Zhang and Cutkosky (2024) demonstrate that a version of composite objective Online Mirror Descent (OMD) (Beck and Teboulle, 2003; Duchi et al., 2010) provides the following regret bound for chosen decreasing step sizes $\eta_t > 0$ (i.e., $0 < \eta_{t+1} \leq \eta_t$ for all $t$):

$$
\sum_{t=1}^{T} [\ell_t^{\mathbf{v}}(\boldsymbol{\delta}_t) - \ell_t^{\mathbf{v}}(\mathbf{u})] \leq \left( \frac{2}{\eta_{T+1}} + \frac{\mu_{T+1}}{2} \right) \|\mathbf{u}\|^2 + \frac{1}{2} \sum_{t=1}^{T} \eta_t \|\mathbf{v}_t\|^2. \tag{34}
$$

See their Theorem 4.1 for details. The version of composite objective Online Mirror Descent (OMD) considered by Zhang and Cutkosky (2024) is initialized with $\boldsymbol{\delta}_1 = \mathbf{0}$ and updated as:

$$
\boldsymbol{\delta}_{t+1} := \arg\min_{\boldsymbol{\delta}} \left[ \langle \mathbf{v}_t, \boldsymbol{\delta} \rangle + \frac{1}{2\eta_t} \|\boldsymbol{\delta} - \boldsymbol{\delta}_t\|^2 + \frac{\mu_{t+1}}{2} \|\boldsymbol{\delta}\|^2 + \frac{\left( \frac{1}{\eta_{t+1}} - \frac{1}{\eta_t} \right)}{2} \|\boldsymbol{\delta}\|^2 \right]. \tag{35}
$$

For more details, see Section 4 therein.

Computing the argmin of the right-hand side, the update rule (35) can be equivalently written as:

$$
\boldsymbol{\delta}_{t+1} := \frac{1}{1 + \eta_t \mu_{t+1} + \eta_t \left( \frac{1}{\eta_{t+1}} - \frac{1}{\eta_t} \right)} (\boldsymbol{\delta}_t - \eta_t \mathbf{v}_t). \tag{36}
$$

Now, with the choice $\mathbf{v}_t := \beta^{-t} \mathbf{g}_t$, $\mu_t = \beta^{-t} \mu$, and $\eta_t = \beta^t \eta$, the update rule (36) becomes:

$$
\boldsymbol{\delta}_{t+1} := \frac{1}{1 + \frac{1}{\beta} \eta \mu + \frac{1}{\beta} - 1} (\boldsymbol{\delta}_t - \eta \mathbf{g}_t), \tag{37}
$$

which, after rearranging, precisely matches ($\beta$-OMD) in the statement of Lemma 3.2.

Moreover, plugging this choice into the regret bound (34), we get:

$$\sum_{t=1}^{T} \langle \beta^{-t} \mathbf{g}_t, \boldsymbol{\delta}_t - \mathbf{u} \rangle \leq \left( \frac{2}{\beta^{T+1}\eta} + \frac{\beta^{-(T+1)}\mu}{2} \right) \|\mathbf{u}\|^2 + \frac{1}{2} \sum_{t=1}^{T} \beta^t \eta \left\| \beta^{-t} \mathbf{g}_t \right\|^2, \tag{38}$$

which, after multiplying both sides by $\beta^T$, becomes:

$$\mathsf{Regret}_T^{[\beta]}(\mathbf{u}) \leq \beta^{-1} \left( \frac{2}{\eta} + \frac{\mu}{2} \right) \|\mathbf{u}\|^2 + \frac{1}{2} \sum_{t=1}^{T} \eta \beta^{T-t} \left\| \mathbf{g}_t \right\|^2. \tag{39}$$

Now let us simplify this into the discounted regret bound in Lemma 3.2. Since we have $\mathbb{E} \left\| \mathbf{g}_t \right\|^2 \leq G^2 + \sigma^2$ for all $t$, the discounted regret bound (39) can be further upper bounded as:

$$\mathbb{E}[\mathsf{Regret}_T^{[\beta]}(\mathbf{u})] \leq \frac{\mu}{2} \frac{\|\mathbf{u}\|^2}{\beta} + \frac{2}{\eta} \frac{\|\mathbf{u}\|^2}{\beta} + \frac{1}{2} \sum_{t=1}^{T} \eta \beta^{T-t}(G^2 + \sigma^2),$$

and

$$\mathbb{E}[\mathsf{Regret}_T^{[\beta]}(\mathbf{u})] \leq \frac{\mu}{2} \frac{\|\mathbf{u}\|^2}{\beta} + \frac{2}{\eta} \frac{\|\mathbf{u}\|^2}{\beta} + \frac{1}{2} \frac{(G^2 + \sigma^2)}{1 - \beta} \eta.$$

Choosing $\eta = \frac{2\|\mathbf{u}\|\sqrt{1-\beta}}{G+\sigma}$ and noticing that the right-hand side no longer depends on $T$, the discounted regret bound becomes:

$$\forall t \in [T], \quad \mathbb{E}\left[ \mathsf{Regret}_t^{[\beta]}(\mathbf{u}) \right] \leq \frac{2 \|\mathbf{u}\| (G + \sigma)}{\beta\sqrt{1 - \beta}} + \frac{\mu}{2} \|\mathbf{u}\|^2.$$

This gives the discounted regret bound given in Lemma 3.2.

## D.2. Proof of Proposition 5.1

From the definition of the $\mathbf{z}$-iterates (8), it holds that

$$\mathbf{z}_{t+1} - \mathbf{z}_t = \left( \mathbf{x}_{t+1} + \frac{1}{1 - \zeta} \boldsymbol{\delta}_{t+1} \right) - \left( \mathbf{x}_t + \frac{1}{1 - \zeta} \boldsymbol{\delta}_t \right)$$

$$= \left( \frac{1}{\zeta} + \frac{1}{1 - \zeta} \right) \boldsymbol{\delta}_{t+1} - \frac{1}{1 - \zeta} \boldsymbol{\delta}_t$$

$$= \frac{1}{1 - \zeta} \left( \left( \frac{1 - \zeta}{\zeta} + 1 \right) \boldsymbol{\delta}_{t+1} - \boldsymbol{\delta}_t \right) = -\frac{\eta}{1 - \zeta} \mathbf{g}_t.$$

Hence, the desired conclusion follows.

## D.3. Proof of Proposition 5.2

From the proof of Corollary 5.1 (see Appendix C), we have the relationship:

$$\eta_\star \mu_\star = \frac{\varepsilon^2}{7(G + \sigma)^2}. \tag{40}$$

Since $\beta_\star = 1 - \left( \frac{\varepsilon}{7(G+\sigma)} \right)^2$, we can deduce the following:

$$1 - \zeta_\star = 1 - \frac{\beta_\star}{1 + \eta_\star \mu_\star} = \frac{(1 - \beta_\star) + \eta_\star \mu_\star}{1 + \eta_\star \mu_\star} = \frac{\Theta\left( \frac{\varepsilon^2}{(G+\sigma)^2} \right)}{1 + \Theta\left( \frac{\varepsilon^2}{(G+\sigma)^2} \right)} = \Theta\left( \frac{\varepsilon^2}{(G + \sigma)^2} \right). \tag{41}$$

From this, the conclusions of Proposition 5.2 follow.

