# OpenReview forum: "General framework for online-to-nonconvex conversion: Schedule-free SGD is also effective for nonconvex optimization"
_ICML.cc/2025/Conference — ICML 2025 oral_

### Official Review · Reviewer_TSfg · 2025-02-28

**Overall Recommendation:** 4

**Summary:**

This work investigates the effectiveness of schedule-free methods in nonconvex optimization. The authors first develop a general framework for online-to-nonconvex conversion, which converts a given online learning algorithm into a nonconvex optimization algorithm. This framework not only recovers existing conversions but also leads to two new conversion schemes. In particular, one of these new conversions corresponds directly to the schedule-free SGD, therefore allowing us to establish its optimal iteration complexity for nonsmooth nonconvex optimization.
The analysis results also provide valuable insights into the parameter choices for schedule-free SGD in practical applications.

**Claims And Evidence:**

Yes

**Essential References Not Discussed:**

No.

**Experimental Designs Or Analyses:**

No experiments.

**Methods And Evaluation Criteria:**

Yes

**Other Comments Or Suggestions:**

Several algorithms (5,6,7) correspond to the same algorithm. It seems that only Algorithm 6 is necessary.

**Other Strengths And Weaknesses:**

This is a solid theory paper on extending the theoretical understanding of schedule-free methods to nonconvex optimization. The main contribution is the development of a general framework for online-to-nonconvex conversion, which converts a given online learning algorithm into a nonconvex optimization algorithm. By specifying the choice of sequences, this framework can recover existing conversions and also leads to two new conversion schemes that cover the schedule-free SGD method. Overall, I found the contribution of this work very fundamental, and the developed framework may spark new research directions on nonconvex optimization algorithm design and analysis.

**Questions For Authors:**

The authors claim that the analysis result can explain why $\kappa_t$ should be chosen close to 1 in practice. I don't see a strong evidence here. The theoretical choice of $\kappa_t \approx 1$ is to achieve theoretical guarantee in general nonconvex optimization, while the practical choice of $\kappa_t \approx 1$ is to achieve the best performance for a specific problem.

**Relation To Broader Scientific Literature:**

May inspire new optimization algorithm design and analysis in nonconvex machine learning.

**Theoretical Claims:**

No.

---

> ### Author Rebuttal · Authors · 2025-03-30
>
> Thank you for the thoughtful question. We agree that a deeper understanding of optimizer behavior ultimately requires incorporating finer-grained properties of the training loss landscape.
>
> That said, a somewhat surprising takeaway from our work—especially in light of your comment—is that certain practical choices of hyperparameters can already be explained using fairly generic assumptions on nonsmooth, nonconvex losses. This suggests that some aspects of real-world training dynamics may be governed by broader principles.
>
> We believe that incorporating additional structural assumptions on the loss—closer to the training losses in practice—could further bridge the gap between theory and empirical behavior, and we view this as a promising direction for future work.

---

### Official Review · Reviewer_upc5 · 2025-03-10

**Overall Recommendation:** 4

**Summary:**

This paper develops a general framework for online-to-nonconvex conversion, which reduces the problem of finding a stationary point of a non-convex objective function to an online learning problem. Their framework extends the work of Zhang & Cutkosky (2024) with a tighter analysis, and it also leads to two new conversion schemes. All three schemes are shown to achieve the optimal convergence rate for nonsmooth, nonconvex stochastic optimization problems. Moreover, the third scheme is shown to have the form of the schedule-free SGD in Defazio et al. (2024). As a result, it complements the convergence analysis for the convex setting in that work and establishes that schedule-free SGD remains optimal in the nonsmooth, nonconvex settings with the proper choice of parameters.

**Claims And Evidence:**

Most claims in this paper are supported by rigorous mathematical proofs. However, I have some reservations regarding the practical insights provided by the analysis.
- The authors observe that under their framework, the choice of $\kappa_t$ for schedule-free SGD is close to 1, which aligns with the empirical observations in Defazio et al. (2024). On the other hand, other aspects of the parameter selection do not exactly match practical choices. Specifically, in Defazio et al. (2024), the choice of $c_t$ is independent of $\kappa_t$ and scales as $1/t$, which is different from the suggestion in Proposition 5.2. Due to this discrepancy, it seems unclear which formulation better explains the empirical success of schedule-free SGD.

- At the end of Section 5, the authors claim that the learning rate $\gamma$ for schedule-free SGD is $\Theta(\frac{(G+\sigma)^2}{\epsilon^2})$ times larger than the optimal step size for SGD with momentum. However, in Proposition 5.2, $\gamma$ is compared to the **OMD step size $\eta_*$** for the online learning problem, which does not directly correspond to the step size in SGD with momentum. In fact, as mentioned by the authors, SGD with momentum corresponds to Option I with OMD as the online learner. Following the reformulation in Zhang and Cutkosky (2024), the effective learning rate for SGD with momentum is given by $\frac{\beta \eta}{\eta \mu + \alpha} = \frac{\eta \xi}{1- \xi}$. Since $\xi$ is close to 1 in the considered setting, the learning rates only differ by a constant close to 1. Hence, the analysis does not necessarily suggest a much larger learning rate than SGD with momentum.

**Essential References Not Discussed:**

I think the authors did a good job and included the most essential references.

**Experimental Designs Or Analyses:**

There are no experiments in this paper.

**Methods And Evaluation Criteria:**

This is a pure theoretical paper, and thus empirical evaluation is not applicable.

**Other Comments Or Suggestions:**

For clarity, it may be helpful to briefly review how Option 1 corresponds to SGD with momentum and contrast it with Option 3. This addition would help readers better understand the differences between these two methods.

Typos:
- Lemma C.1 (Line 645): the denominator in the first term should be $DT$.
- Lemma C.5: On Line 820, the second expectation should be over $\mathbf{X}_{\tau}$. Also, in the proof on Line 832, the left-hand side should be $\\|y_s - x_s\\|$ instead of $\\|w_s-x_s\\|$.

**Other Strengths And Weaknesses:**

The paper is well-written and easy to follow. However, one potential concern is that its theoretical contribution may not be significantly more substantial than that of Zhang and Cutkosky (2024), as it builds upon the same key concepts and techniques. While the authors introduce two new conversion schemes within their new framework, all proposed schemes ultimately achieve the same convergence guarantees (up to a constant). This raises the question of what advantages the newly proposed schemes offer beyond the existing approach.

**Questions For Authors:**

I do not have additional questions.

**Relation To Broader Scientific Literature:**

Two prior works are most relevant to this submission:
- The proposed framework can be viewed as an extension to Zhang and Cutkosky (2024). In particular, it adopts the same key concepts (e.g., the approximate Goldstein stationary point and discounted regret) and the key techniques (e.g., the composite objective OMD). In some sense, the current submission presents the analysis from Zhang and Cutkosky (2024) in a more modular way and make the observation that there is some flexibility in choosing the sequence $\{x_t\}$ in the update rule.
- The main contribution of this paper is to establish the optimal convergence guarantees for schedule-free SGD (Defazio et al., 2024), a recently proposed optimizer with strong empirical performance, in nonsmooth and nonconvex optimization. The original paper focuses on the stochastic convex optimization setting with a completely different analysis, and this paper offers a new perspective on the algorithm's design.

**Theoretical Claims:**

Yes, I have checked the proofs in the Appendix, and to the best of my knowledge they are correct, except for some minor typos.

---

> ### Author Rebuttal · Authors · 2025-03-30
>
> Thank you for your constructive comments!
>
> **Regarding the step size comparison.**
>   Thank you for pointing this out. Indeed, the step size of our Schedule-Free algorithm should not be compared with the step size of the OMD, and instead with the ``effective'' step size of the momentum method of Zhang and Cutkosky (2024). We agree with the observation that the stepsizes are of the same order and will update the presentation accordingly.
>
>
>  **Regarding $\kappa_t$.** We acknowledge that our nonconvex analysis adopts a different averaging scheme, using $c_t = 1 - \zeta$ instead of the original choice $c_t = \frac{1}{t}$ proposed by Defazio et al. (2024). A more detailed investigation of this difference is an interesting direction for future work, and we will clarify this point in the final version of the paper.
>
> That said, our preliminary experiments suggest that the specific choice of $c_t$ has limited impact on the performance of schedule-free methods. For example, when training a ResNet on CIFAR-10, we often find that EMA averaging (as used in our method) results in more stable training compared to the full averaging scheme of Defazio et al. (2024). Here is the link for the experiments https://docs.google.com/document/d/1WBeV1DuS_zTZ6370hfRvQk617NILsp5uWwvD1CYUn74/edit?tab=t.0 .
>
>
> In contrast, the choice of $\kappa_t$ plays a much more critical role in practice. Performance degrades significantly when $\kappa_t$ is set below $0.9$, highlighting the importance of tuning this parameter carefully. Unlike the convex analysis of Defazio et al. (2024), which allows $\kappa_t$ to be chosen arbitrarily ($\beta_t$ in their notation), our nonconvex analysis imposes a more realistic constraint and, in this sense, better aligns with the practical settings.
>
>
> **Typos and other suggestions.** Thanks for the suggestions regarding the presentation and pointing out the typos. We will update the paper accordingly.

---

> > ### Comment · Reviewer_upc5 · 2025-04-03
> >
> > I thank the authors for their detailed response and the additional experiment. As my concerns have been fully addressed, I am happy to raise my score to 4. I encourage the authors to incorporate the clarifications provided into the revision.

---

### Official Review · Reviewer_y9cm · 2025-03-13

**Overall Recommendation:** 4

**Summary:**

This paper introduces a more general online-to-nonconvex reduction. Based on the an OMD variant with discounted regret guarantees, the optimal convergence rate to a $(\lambda,\delta)$-stationary point is shown for three different variants. The third variant is shown to coincide with schedule-free SGD algorithm with specific parameter choices. The parameter choices required to achieve optimal rates for this variant of schedule-free SGD correspond to those that lead to the best empirical performance, which was not explained by the theory of schedule-free SGD in the convex setting.

**Claims And Evidence:**

Yes.

**Essential References Not Discussed:**

No

**Experimental Designs Or Analyses:**

-

**Methods And Evaluation Criteria:**

-

**Other Comments Or Suggestions:**

-

**Other Strengths And Weaknesses:**

Designing and analyzing algorithms for the nonconvex and non-smooth setting is a very interesting direction. Given that SFO works so well in practice, it is very interesting to see that it is optimal for the nonconvex and non-smooth setting and that the parameter choices required to achieve optimal rates for this variant of schedule-free SGD correspond to those that lead to the best empirical performance.

**Questions For Authors:**

-

**Relation To Broader Scientific Literature:**

The key contribution is the online-to-nonconvex conversion framework which is an extension of the work of Cutkosky et al. (2023). Using this reduction with a specific version of OMD, the authors show that the resulting algorithm corresponds to schedule-free SGD (Defazio et al., 2024), which was only studied in the convex setting before, and achieves  optimal worst-case convergence rates.

**Theoretical Claims:**

No.

---

> ### Author Rebuttal · Authors · 2025-03-30
>
> Thank you for encouraging feedback!
> We do agree that designing and analyzing algorithms for the nonconvex and non-smooth setting is a very interesting direction.
> We also agree that it's nice to see have strong theoretical guarantees for practical optimizers.

---

### Official Review · Reviewer_vDmv · 2025-03-14

**Overall Recommendation:** 4

**Summary:**

This paper presents a general framework for conversion of any online-learning algorithm into a non-convex (non-smooth) optimization algorithm. The authors provide general non-convex convergence guarantees for the online-to-nonconvex unified framework in terms of the $(\lambda, \epsilon)$-stationary point. Their analysis, which applies to any online algorithm with appropriate discounted regret guarantees, is subsequently refined for the case of the discounted online mirror descent algorithm. Furthermore, the authors present three different conversions schemes as special cases of their framework. They show that their last conversion enables the recovery of the schedule-free SGD method.

## Update after rebuttal:
I thank the authors for their response to my questions. Since my recommendation is already for acceptance, I will keep my score.

**Claims And Evidence:**

All the claims made in this paper are supported by sufficient evidence.

**Essential References Not Discussed:**

No.

**Experimental Designs Or Analyses:**

Not applicable.

**Methods And Evaluation Criteria:**

The methods and evaluation criteria are appropriate for the given problems.

**Other Comments Or Suggestions:**

No.

**Other Strengths And Weaknesses:**

Strengths:
- This paper follows a good structure, the writing is good and most claims are presented with sufficient explanations to help explain the main results.
- The designed random EMA iterate is a novel scheme that allows for improved convergence results and is consistent with practical applications.
- The convergence analysis provided in this work is solid and based on relatively mild assumptions.

Weaknesses:
- I think including some experimental results possibly in some simple settings could help illustrate the behavior of the presented algorithms and potentially help clarify the advantage of schedule-free methods for non-convex optimization.

**Questions For Authors:**

1. Could the authors give some intuition/explanation about the choice of the comparators for the online learner in Lemma 3.1?
2. Could the authors elaborate a bit more on the motivation and intuition for the iterates produced by option II and the anchoring scheme?

**Relation To Broader Scientific Literature:**

This work extends the online-to-nonconvex framework to recover some popular optimizers, which have been shown to have optimal guarantees. This idea provides insight into the success of these algorithms even for the non-convex non-smooth case and allows for a better understanding of the step size choices of these in practice. This paper also introduces a new random EMA scheme, which is used for the algorithmic output. This method allows (slightly) improved convergence guarantees as compared to previous works selecting the output uniformly at random.

**Theoretical Claims:**

I looked through the proofs in Appendix C and D and they seem to be correct.

---

> ### Author Rebuttal · Authors · 2025-03-30
>
> Thank you for your feedback and suggestions.
>
>
> - The choice of these comparator sequences can be motivated by viewing them as the "good" update direction in hindsight. If we ignore EMA averaging in the choice of comparators (for simplicity), we can see that the comparator points exactly in the direction of $\nabla F(y_t)$, i.e., the ``lookahead'' gradient. Notice that at iteration $t-1$, we do not have access to this gradient, and that's why we call this lookahead.
>
>
>
>
> - As can be seen from the algorithm, each epoch can be viewed as performing local optimization around an anchoring point. However, if the algorithm spends too much time around the same anchor point, the landscape may remain underexplored, preventing the algorithm from reaching the desired stationary point. Therefore, once the algorithm completes an epoch, it shifts away to explore other regions of the landscape.
>      As mentioned in the paper, this algorithm design has a nice connection to previous nonconvex optimization approaches that repeatedly solve convex subproblems constructed via appropriate regularization (e.g., Chen and Hazan (2024)).

---

### Decision · Program_Chairs · 2025-05-01

**Decision:**

Accept (oral)

**Comment:**

This work develops a general framework for online-to-nonconvex optimization, which converts a given online learning algorithm into a nonconvex optimization method with strong guarantees. Specifically, the proposed framework not only recovers existing conversion schemes but also leads to a couple of new ones, from which a schedule-free SGD method emerges as an output. All the reviewers unanimously agree with the acceptance of the paper and believe that the paper is with rigorous and transparent analysis.